# Bumps in river profiles: uncertainty assessment and smoothing using quantile regression techniques

Wolfgang Schwanghart[1], Dirk Scherler[2,3]

[1] University of Potsdam, Institute of Earth and Environmental Science, 14476 Potsdam-Golm, Germany
[2] GFZ German Research Centre for Geosciences, Earth Surface Geochemistry, 14473 Potsdam, Germany
[3] Freie Universität Berlin, Institute for Geological Sciences, 12249, Berlin, Germany

*Correspondence to*: Wolfgang Schwanghart (w.schwanghart@geo.uni-potsdam.de)

**Abstract.** The analysis of longitudinal river profiles is an important tool for studying landscape evolution. However, characterizing river profiles based on digital elevation models (DEM) suffers from errors and artifacts that particularly prevail along valley bottoms. The aim of this study is to characterize uncertainties that arise from the analysis of river profiles derived from different, near-globally available DEMs. We devised new algorithms – quantile carving and the CRS algorithm – that rely on quantile regression to enable hydrological correction and uncertainty quantification of river profiles. We find that globally available DEMs commonly overestimate river elevations in steep topography. The distributions of elevation errors become increasingly wider and right-skewed if adjacent hillslope gradients are steep. Our analysis indicates that the AW3D DEM has the highest precision and lowest bias for the analysis of river profiles in mountainous topography. The new 12-m resolution TanDEM-X DEM has a very low precision, most likely due to the combined effect of steep valley-walls and the presence of water surfaces in valley bottoms. Compared to the conventional approaches of carving and filling, we find that our new approach is able to reduce the elevation bias and errors in longitudinal river profiles.

## 1 Introduction

Rivers play a dominant role in the topographic evolution of the Earth surface, and possibly other planetary bodies (Hack, 1957; Howard, 1998; Whipple et al., 2013). They transfer sediment from mountains to depositional basins, set the base level for hillslopes, and convey tectonic and climatic signals across landscapes. The geometry of a river and specifically its longitudinal profile, reflect the climatic and tectonic forcing, variations in base level and sediment transport processes as well as differences in bedrock erodibility. Gradients along rivers, for example, reflect spatial variations in uplift rates (Whipple et al., 2013; Mudd et al., 2014; Scherler et al., 2014) and indicate the extent of past glaciations (Brardinoni and Hassan, 2006). Moreover, they can act as predictors for the zones of erosion and sediment accumulation during extreme events (Devrani et al., 2015) and reflect the repeated impact of masswasting events (Korup, 2006). Longitudinal river profiles and metrics derived from them (e.g., the normalized channel steepness metric $k_{sn}$ (Wobus et al., 2006)) have become

important tools for studying the topographic evolution of mountain belts and deciphering changes in climate and tectonics (Bishop et al., 2005).

Data on longitudinal river profiles are usually derived from digital elevation models (DEMs) (Gonga-Saholiariliva et al., 2011; Wobus et al., 2006). Different sensors and techniques such as radar imaging, stereoscopy of optical imagery, terrestrial and airborne laser scanning, and structure-from-motion (SfM) provide DEMs at ever-increasing spatial resolution with unprecedented coverage. However, DEMs contain errors due to uncertainties in data acquisition, interpolation techniques and spatial resolution (Fisher and Tate, 2006; Purinton and Bookhagen, 2017). Structures such as bridges, culverts and reservoirs affect longitudinal river profiles derived from DEMs in ways that can either hide features present in reality or introduce patterns that do not represent the actual course of the profile (Schwanghart et al., 2013). Deviations from graded river profiles may reflect signals of climatic or tectonic perturbations, but often they relate to errors and artifacts that generate bumpy river profiles and thus introduce uncertainties that may compromise the interpretation of longitudinal river profiles (Harbor et al., 2005; Hayakawa and Oguchi, 2006; Wobus et al., 2006).

The objective of this study is to characterize and quantify the uncertainties of elevation values in longitudinal river profiles derived from near-globally and publicly available DEMs including the new TanDEM-X DEM (Wessel, 2016). To attain this goal, we devise new algorithms (quantile carving and constrained regularized smoothing) that use non-parametric quantile regression for assessing uncertainties and smoothing of river profiles. Using LiDAR DEMs as benchmark data and our new algorithms, we study how longitudinal river profiles derived from globally available DEMs (Table 1) are affected by errors and how these errors depend on the topographic setting. Moreover, we examine the best choice of parameter values for our algorithms to guide their application. Our algorithms will aid the visual interpretation and automated analysis of longitudinal river profiles and have additional applications in hydrodynamic modelling.

## 2 DEM artifacts and the longitudinal river profile

DEMs are the product of a processing chain that converts spatially discrete altitude measurements into a simplified representation of the Earth surface (Wechsler and Kroll, 2006). The processing chain typically involves ground-based, airborne, or satellite-based data acquisition, post-processing, and georeferencing. Subsequent interpolation and filtering operations usually convert these measurements into gridded datasets (Fisher and Tate, 2006). Because each processing step contains assumptions and analytical uncertainties, DEMs are not perfect representations of the land surface but contain errors that propagate through DEM-derived products (Fisher and Tate, 2006; Holmes et al., 2000; Reuter et al., 2009; Schwanghart and Heckmann, 2012).

DEM errors are either random or systematic. Random errors typically occur due to uncertainties related to instrument precision. Radar-derived DEMs, for example, include speckle and thermal noise as well as timing and positioning errors (Falorni et al., 2005; Rizzoli et al., 2012). Random errors may also be autocorrelated, i.e., they cluster spatially (Oksanen and Sarjakoski, 2006). Systematic errors are specific to the type of acquisition method. Lens distortions may affect DEMs

generated from both photogrammetric and SfM methods (James and Robson, 2014). Foreshortening, layover, and shadowing produce errors and voids in radar-derived DEMs (Falorni et al., 2005). Systematic errors are also found in DEMs interpolated from topographic contour lines (Guth, 1999; Wobus et al., 2006). While random errors tend to hide deterministic structure in the data, systematic errors may lead to wrong interpretations inferred from structures not present in reality.

Providers of DEM data usually describe the accuracy of their product through comparison with ground control points. More detailed error analyses, however, reveal the spatial variability of height errors in DEMs (Gorokhovich and Voustianiouk, 2006; Scherler et al., 2008). Global accuracy statistics are thus often insufficient to inform about the uncertainties of DEMs (Carlisle, 2005) and may lead to grave underestimation of the uncertainties in DEM-based modelling results (Canters et al., 2002; Hancock et al., 2006; Nardi et al., 2008).

Rivers present only a minor portion of the cells in a DEM, but this portion is particularly prone to certain types of errors (Guth, 2006; McMaster, 2002). Elevation anomalies along valley bottoms of the Pieniny Mountains, Poland, were found to range from -39 m to +145 m in the X-band SRTM and in the ASTER GDEM from -52 m to +88 m (Czubski et al., 2013; Ludwig and Schneider, 2006), thus vastly exceeding reported DEM accuracies (Jarvis et al., 2008; Tachikawa et al., 2011). Low DEM accuracy along rivers and valley bottoms can have several causes. In steep terrain, shallow viewing angles of radar systems can lead to shadow-related data loss. Although this problem can be mitigated by combining data from ascending and descending paths, it often cannot be completely eliminated (e.g., Rabus et al., 2003), and leads to systematically higher valley bottom altitudes in the C-Band SRTM DEM compared to the ASTER GDEM (Hayakawa et al., 2008). Difficulties in penetrating through thick riparian vegetation cause additional problems with DEMs derived from radar systems (Baade and Schmullius, 2014) and optical imagery (Gesch et al., 2012). As a consequence, patches of dense floodplain vegetation are likely to increase DEM values along river reaches, in particular where they are narrow. Although LiDAR derived DEMs often comprise benchmark data in terms of accuracy and resolution, gridded elevations along valley bottoms may still contain errors due to specular reflection and signal absorption by water (Malinowski et al., 2016).

In addition, DEM resolution limits the accurate representation of valley bottom heights. Where valley bottom width is less than twice the pixel size, DEMs may not capture the actual cross sectional shape correctly (Hengl, 2006). Finally, DEMs are simply unable to represent embankment underpasses such as bridges or culverts (Lindsay and Dhun, 2015). Unless manually or automatically removed, these features appear as positive excursions in the longitudinal river profiles, particularly at high spatial resolution (LiDAR-derived DEMs) and in human-altered landscapes (Schwanghart et al., 2013; Sofia et al., 2013).

Accurate representation of drainage structures can be obtained by algorithms that interpolate DEMs from scattered data points and contour lines (Hutchinson et al., 2011). Most of the above-mentioned errors, however, cannot be avoided during DEM generation and require post-processing steps. Flow path enforcement (Lindsay, 2016) is a class of algorithms to establish flow connectivity between all pixels within a DEM and its edges. While these algorithms primarily adjust planform flow patterns, they can also be used to modify DEM values. For example, *flood filling* of sinks ensures that each pixel has at least one neighboring pixel with equal or lower elevation (Fig. 1a). The drawback of this approach is that it creates

potentially large flat areas where topographic information is lost, and that it leads to unrealistic flow patterns imposed on the filled topography (Soille et al., 2003) that ultimately affect longitudinal river profiles (Watson et al., 2015). In contrast, *carving* (Fig. 1b), is a procedure to cut through blockages, typically by imposing the constraint that while moving downstream, no pixel is higher than its upstream neighbors (Lindsay, 2016; Schwanghart et al., 2013; Soille et al., 2003). In

addition, there exist combinations of carving and filling that minimize the costs (i.e., measured by the total amount of DEM modifications) of transforming the DEM during flow enforcement (Lindsay, 2016; Soille, 2004). However, no matter which flow enforcement approach is used, the derived river profiles will most certainly contain zero-gradient sections (Nardi et al., 2008) with abrupt steps at their downstream facing side. Distinguishing such "artificial riffle-pool sequences" from actual ones is often not easy (Hayakawa and Oguchi, 2006).

Common approaches for removing errors in river profiles include running average filters (Hayakawa and Oguchi, 2006), contour-interval subsampling (Wobus et al., 2006), smoothing splines (Harbor et al., 2005), and locally weighted regression (Aiken and Brierley, 2013). However, all of these approaches face similar problems: they fail to remove spikes if the length of the moving window is too small and average out potentially important features if the window is too large (Aiken and Brierley, 2013). Furthermore, conventional smoothing algorithms are typically applied to individual river reaches rather than

entire river networks, which may lead to sharp discontinuities at river confluences. Finally, even robust approaches that aim at reducing the influence of individual outliers on the smoothed curve (Aiken and Brierley, 2013) may yield profiles that contain sections with downstream increasing elevation that result from long sections where valley bottom elevations are overestimated. An example of how outliers may generate downstream increasing elevations in profiles smoothed by a running average is shown in Fig. 1c. Harbor et al. (2005) used an iterative procedure that lowers downstream locations and

raises upstream locations to correct such sections but thus produced river profiles may not be consistent with the data.

What is thus needed is an approach that (i) filters DEM artifacts from longitudinal river profiles while ensuring downstream decreasing elevations, and that (ii) accounts for the fact that river elevations are usually overestimated. Moreover, the approach should be applicable to entire river networks instead of single rivers only, and should be flexible enough to deal with breaks in along-river slopes where these actually exist.

**3 The CRS algorithm**

We wish to transform a bumpy river profile into a smoother profile so that it ideally contains elevations and along-river gradients that approximate those in reality. Common approaches such as the running average and local regression – possibly combined with flow enforcement techniques – insufficiently cope with systematic biases and outliers, and may even produce profiles that increase in downstream direction (Fig. 1). Our approach avoids these drawbacks and combines flow

enforcement and smoothing in one procedure.

We developed a non-parametric quantile regression that we term CRS (constrained regularized smoothing) algorithm. Our approach exploits the network topology of river networks (see Appendix A1) and estimates the $\tau$th quantile function $Q_{Z|X}(\tau)$

of the random variate elevation $Z$ conditional on the distance $X$ in the upstream direction from the outlet. We chose a quantile regression rather than a least squares approach (see Appendix A2) despite the computational appeal of least squares. Quantile regression makes no assumptions about the underlying probability distribution of the observational noise and is more robust to outliers. Moreover, quantile regression provides the opportunity to obtain a comprehensive estimate of the distribution of $Z$ (Koenker, 2010) and is therefore suited to applications that are interested in more than the mean as a single sample.

We propose a non-parametric approach to regression so that the river profile does not take a predetermined shape. River profiles are usually too variable to be characterized by a functional model (e.g. the stream power model) and any strict assumptions about profile form, e.g. a concave upward steady state profile, would limit the range of potential profiles the algorithm could be applied to (Shepherd, 1985). In turn, a non-parametric free-form solution entails the inference of a very large number of parameters. In fact, there are as many parameters as elevation values in the profile, such that the problem is ill-posed. Regularization refers to the approach to include additional constraints to the solution of the problem. Here we use regularization to encode preference for local smoothness and spatial autocorrelation (Sivia and Skilling, 2006) of the river profile, an assumption that is vital for any kind of spatial analysis (De Smith et al., 2007). Given that there is usually only one realization of $Z$ for each value of $X$, assumptions about local smoothness are indeed necessary to make any statements about the distribution of $Z(X)$. However, this requirement can be relaxed if we force profiles to decrease in downstream direction. Here we first show how such a monotonicity constraint leads to an approach that we term *quantile carving* before we proceed with the CRS algorithm.

Commonly, quantiles are derived by sorting data and determining the values that separate the data into the desired proportions. However, quantiles can also be defined as an optimization problem of minimizing a sum of symmetrically (in case of the median) or asymmetrically weighted absolute residuals (Koenker, 2010). If the $\tau$th quantile function is $Q_z(\tau) = Iz_\tau$ with I being the identity matrix, then $z_\tau$ is found by minimizing the argument

$$\arg\min \sum_{i=1}^{n} \left( \rho_\tau (z - Iz_\tau) \right) \tag{1}$$

where $\rho_\tau$ is the loss function

$$\rho_\tau (z - Iz_\tau) = (z - Iz_\tau)(\tau - \mathbb{I}_{(z-Iz_\tau)<0}) \tag{2}$$

and $\mathbb{I}$ is an indicator function that has a value 1 if $(z - Iz_\tau) < 0$ and 0 otherwise (Koenker, 2010). At this point, however, $z_\tau$ is not conditional on distance x, but calculated independently for each point along the river profile. Unless multiple samples exist for each location, which is usually not the case, $z_\tau$ equals $z$. However, we can explicitly condition $z_\tau$ by adding prior knowledge about river geometry, i.e., by constraining elevations to be monotonously decreasing downstream ($z_\tau(x)$ is greater or equal than any downstream elevation). Formally, this is achieved by minimizing eq. (1) subject to the inequality constraint $z_\tau(x) \geq z_\tau(x - \delta x)$ which is a minimization problem that can be solved by linear programming methods

(Koenker, 2010) (see Appendix A3). We refer to this approach of processing the longitudinal river profile as *quantile carving* and show examples of its application in Fig. 2. For quantiles approaching 0 and 100%, the resulting profiles will follow those of the commonly used hydrological conditioning approaches carving and filling, respectively. Thus, carving and filling are extremal approaches and quantile carving provides a statistical framework that links the two.

A disadvantage of the commonly applied techniques of carving and filling is that they often introduce zero-gradient sections in the profile that are typically separated by artificial steps (Fig. 1). The same applies to quantile carving and disrupts our aim of a realistic representation of the river profile (Fig. 2). A smoother profile is characterized by less local variability and is devoid of sharp kinks. Given a longitudinal river profile, there are many ways of measuring how rough or wiggly the profile is. An intuitively appealing measure is the integrated second derivative $\int [z_\tau''(x)]^2 dx$ which is not affected by the

addition of a constant or linear function and has considerable computational advantages (Green et al., 1993). The CRS algorithm expands on quantile carving by determining the cost of a particular solution not only by its goodness-of-fit, but also by its roughness. This entails additionally minimizing the integrated second derivative so that the optimization involves a quadratic problem:

$$\arg\min \sum_{i=1}^{n} \left( \rho_\tau(z(x) - Iz_\tau(x)) \right) + s \int [z_\tau''(x)]^2 dx \tag{3}$$

$$\text{subject to } z_\tau(x) \geq z_\tau(x - \delta x)$$

The scalar $s$ determines the weight of the integrated second derivative and is calculated by Eq. (4) where $\Delta x$ is the spatial resolution of the DEM, $n$ is the number of nodes in the network and $p$ is the number of second derivative equations:

$$s = (\Delta x)^2 K \sqrt{\frac{n}{p}} \tag{4}$$

Larger values of the smoothing parameter $K$ result in smoother profiles (Fig. 3). We include the squared spatial resolution into the equation so that any linear transformation (e.g. distance scaling) of the river profile does not affect the smoothing results. Yet, this entails that profiles with different spatial resolutions must be smoothed with different values of $K$ to obtain

the same or at least similar results. If two profiles A and B have different spatial resolutions $\Delta x_A$ and $\Delta x_B$, then smoothing both profiles will return similar results if the smoothing parameter $K_B$ is calculated from $K_A$ by Eq. (5).

$$K_B = \left( \frac{\Delta x_A}{\Delta x_B} \right)^2 K_A \tag{5}$$

The quadratic term in Eq. (3) renders the optimization problem suitable for quadratic programming (Takeuchi et al., 2005). By formulating the CRS algorithm as a quadratic programming problem (see Appendix A4), we obtain high flexibility for setting and relaxing constraints on the resultant profiles. For example, we may want to force elevations at particular locations

to take on predefined values that were measured in the field and are precisely known. In other locations such as waterfalls and rapids, dams, or river confluences, profiles may actually exhibit high curvatures that should remain in the smoothed profile. Provided that these locations are precisely known, the curvature constraint can be locally relaxed or entirely removed by setting variable values of $K$. Thus, unlike previous approaches, the CRS algorithm allows for incorporating prior knowledge about river-profile geometry and independent observations.

Longitudinal river profiles of the Big Tujunga catchment (San Gabriel Mountains, USA) and derived from the SRTM-1 are shown in Fig. 3. The profile values show large fluctuations that are particularly high in the central part of the profile where hillslope gradients adjacent to the river are highest. Results of the CRS algorithm for different values of $K$ and $\tau$ are shown in Fig. 3b-e. Thus derived profiles are monotonously decreasing downstream while filtering the wiggles. The amount of smoothing is dictated by $K$. For $K=1$ and $\tau = 0.5$, the profile contains various steps and these steps remain for different values of $\tau$. A higher value of $K$ more effectively filters the profile, but strongly influences the range between the $10^{th}$ and $90^{th}$ percentiles at some locations. The tributary in Fig. 3d and e lacks the small scale variability such that the interquantile range derived with $K=1$ is very low compared to $K=10$. Any uncertainty estimate based on the interquartile range must thus depend on $K$.

## 4 Methods and data

Our goal in this study was to (1) assess the quality of near-globally and publicly available DEMs (hereafter termed global DEMs) for the analysis of longitudinal river profiles, and to (2) examine the best choice of parameter values of the CRS algorithm and compare its performance with commonly used approaches of filling and carving. This allows us to report ranges of parameter values that will guide the application of the CRS algorithm in other areas.

We addressed the first point by comparing river profiles obtained from global DEMs (SRTM-1, ALOS World 3D30 (AW3D), ASTER GDEM v2, TanDEM-X DEM) (see Table 1, Fig. 4). Our test sites are the Yakima catchment in the undulating to hilly landscape of the Washington Mountains, USA, the Big Tujunga catchment in the hilly to mountainous landscape of the San Gabriel Mountains, USA, and the Kali Gandaki catchment in the mountainous landscape of the High and Lesser Himalayas, Nepal. The TanDEM-X DEM is available to us only for the Kali Gandaki catchment. We first compared the global DEM with those of LiDAR DEMs, where available, to obtain estimates of absolute errors. The LiDAR DEMs are bare-earth DEMs derived from point clouds (>3 points/m$^2$) and were downloaded from the OpenTopography facility (Table 1). Due to their decimeter to sub-decimeter accuracy, we consider the LiDAR data as our benchmark DEMs. The acquisition dates of the DEMs vary but we expect that temporal changes in the land surface are minor. We used the reference ellipsoid defined by the world geodetic system WGS 84 as basis for all DEMs and resampled them to the same spatial extent and resolution (30 m) using bilinear interpolation. We then derived flow directions and stream networks for selected channel heads from the downsampled LiDAR DEMs, and obtained elevations along those networks from the global DEMs. This approach may overestimate errors in DEMs due to resampling and because lateral patterns of stream networks

derived from different DEMs may differ. However, 2D crosscorrelation of global DEMs with the LiDAR DEMs did not reveal any systematic horizontal shifts between the DEMs. Due to the lack of benchmark data, we could not quantify absolute errors for the Kali Gandaki. Thus, we next determined relative errors (precision) along the river profiles by smoothing the profiles with the CRS algorithm. We chose the median ($\tau = 0.5$) to obtain profiles along the local central tendency of the data and set the smoothing parameter $K=5$ which we found by visual analysis suitable to filter the data while sufficiently reproducing the structure of the profiles. We quantified the precision using the deviations of the profiles from the CRS-smoothed profiles. Because the study site in Nepal represents a cross-section through the Himalaya with pronounced differences in topographic relief, we also assessed the spatial variability of precision as moving down the river. Specifically, we tested whether median hillslope gradients within a distance of 1000 m of the Kali Gandaki exert any influence on the precision of river profiles.

To address the second point of our study, we used a derivative-free simplex search method to find values of the parameters $K$ and $\tau$ that yield CRS-smoothed profiles that best approximate the LiDAR-derived profiles. Since a good agreement of profiles is largely determined by both similar elevations and gradients, we defined the objective function $f$ to be minimized as a function of the elevation ($z_{i\Delta}$) and gradient offsets ($g_{i\Delta}$) between the profiles

$$f(z_{i\Delta}, g_{i\Delta}) = \sum_i (z_{i\Delta} g_{i\Delta})^2 \tag{6}$$

Again, we have to restrict our analysis to the Yakima and Big Tujunga catchments due to the unavailability of benchmark data for the Kali Gandaki catchment.

Finally, we analyzed the sensitivity of profiles derived by the CRS algorithm to changes of the values $K$ and $\tau$. Specifically, we studied how variations of both parameters affect the goodness of fit between pixel-by-pixel elevations as well as along-river gradients derived from the smoothed profiles and the LiDAR benchmark DEMs.

We implemented the proposed algorithms in TopoToolbox 2 (see also Appendix B), which is software for the analysis of DEMs (Schwanghart and Scherler, 2014). The software is written in MATLAB and thus has direct access to a numerical library of optimization routines usually lacking in standard GIS software. To minimize equations (1) and (3), we used the function linprog and quadprog of the MATLAB optimization toolbox which solve linear and quadratic problems, respectively (see also Appendix A1-4).

## 5 Results

Elevation offsets between the global DEMs and the LiDAR data differ in-between sites and DEM types (Fig. 5). All river profiles are consistently biased towards higher altitudes compared to the LiDAR data (Table 2), but more so in the steeper terrain of the Big Tujunga catchment compared to the Yakima dataset. The observed bias does not affect the entire DEM but approaches zero on hillslopes (Fig. 6) so that we infer that low accuracy particularly affects valley bottoms. Based on kernel-derived probability density distributions, the quantiles of the zero offset are less than the median (except for AW3D in

Yakima), and again lower in steep terrain. Despite predominant overestimation, negative offsets also exist and have minimum values that range from -3 to -45 m. We observe maximum absolute deviations of up to 49 m that are usually positive as indicated by positive skewness values for the SRTM-1 and AW3D datasets. An exception is the ASTER GDEM where offsets are inconsistently skewed towards positive and negative values. In general, the differences between the ASTER GDEM-derived and LiDAR-derived profiles are highly variable as indicated by a high standard deviation in both sites.

Deviations between the original and CRS-smoothed profiles from global DEMs further highlight errors in profiles obtained from different DEMs at different sites, including the site in Central Nepal where LiDAR data is unavailable (Fig. 7). With increasing topographic relief, residuals from the median profile are increasingly biased towards positive values (Table 3). At the same time, the higher root mean squared error (RMSE) shows that increasing relief results in a reduced precision. Elevation values from the AW3D dataset show the highest precision at all sites whereas the TanDEM-X DEM has the highest RMSE (Table 3, Fig. 8). Note that we only took those elevation values from the TanDEM-X DEM into account that are flagged to be consistent, as indicated in the consistency mask that is provided together with the DEM data (Wessel, 2016).

The above observations beg the question of what topographic parameters control the precision of longitudinal river profiles. Our data from Central Nepal show that error magnitudes, estimated as the 95-5% interquantile range, vary spatially (Fig. 9). In particular, there exist sections along the Kali Gandaki River where elevations derived from all DEMs have low precision (Fig. 9a, b). Hillslope gradients within 1000 m distance from the river, measured and averaged within 10-km-long river reaches, correlate with precision (Fig. 9c). This indicates that all DEMs have problems to precisely represent the bottom of valleys straddled by steep canyon walls. Again, the precision differs strongly between the DEMs with ASTER GDEM and TanDEM-X DEM having the lowest precision.

Based on the LiDAR data from the Yakima and Big Tujunga catchments, we found different optimal values of $K$ and $\tau$ for the different DEMs (Table 4). Values of $K$ and $\tau$ are particularly large for the ASTER GDEM, which is quite noisy along the Yakima River. $\tau$ values are always less than 0.5 and lower in the Big Tujunga which is consistent with the percentile value $\tau_0$ (0.01-0.55) determined before (Table 2). In general, the CRS algorithm reduces the deviations from the benchmark profile by lowering the RMSE by a factor of 1.4-2.0. Compared to profiles derived from the carving and filling algorithms, CRS algorithm yields considerable lower RMSE.

The improved agreement between the profiles derived from the CRS algorithm is also reflected by a higher pixel-by-pixel correlation of gradient of CRS derived profiles and those of the benchmark DEMs. Fig. 10 shows a comparison of gradients derived from the global DEMs and preprocessed by different methods for the Yakima catchment. Profiles without preprocessing exhibit the largest scatter. The scatter is most effectively decreased by the CRS algorithm as it reduces the number of zero gradients and artificial steps as compared to the methods of filling and carving.

Figure 11 illustrates the results from a sensitivity analysis of the CRS-derived profiles to changes in K and τ for the Yakima catchment and the AW3D DEM. The sensitivity analysis shows that increasing values of K can excessively smooth the

profile and thus impair the agreement between the profile elevations and gradients. Very low (<0.01) and high (>.099) values of τ can also lead to grave mismatches of the smoothed profile and the benchmark profile as the quantile approach fails to penalize large deviations as opposed to a least-squares approach. The analysis also shows that an agreement between elevations or slopes is not necessarily derived from the same set of parameter values. Whereas an agreement of elevations is largely obtained by varying τ in a rather broad range of K between 0.1 to 3 (Fig. 11a), an agreement of gradients is largely within a narrow range of K and a broad range of τ between 0.3 and 0.8 (Fig. 11b). Fitting profiles to benchmark data may thus significantly depend on whether the objective function minimizes the differences between river elevations or gradients (Fig. 11c).

## 6 Discussion

### 6.1 Positive bias of elevation in river profiles

Different methods of data acquisition and DEM generation have in common that DEM errors along valley bottoms are usually biased towards positive values (Hayakawa and Oguchi, 2006). Our analysis shows that this bias is more pronounced in steep terrain, which calls for caution when interpreting longitudinal river profiles in mountainous landscapes. The bias may at least partly be due to the coarse resolution that averages over finer-scale topographic variability. As we resampled our benchmark LiDAR DEM to the resolution of our tested DEMs and still large differences to global DEMs exist, additional causes of the observed bias must exist that relate to the type of data acquisition and field site characteristics (Vaze et al., 2010). While we cannot resolve the underlying causes in detail, our results suggest that biases affect all DEMs, although to different degrees.

### 6.2 Precision of river profiles derived from global DEMs

In addition to the bias, the distribution of elevation errors in the longitudinal river profiles becomes increasingly wider and right-skewed in landscapes of higher relief. As relief increases, the profiles obtained from all global DEMs have decreasing precision, but there is a more than five-fold difference between the precision of the different DEMs. In all three sites, the AW3D dataset has the highest precision, followed by the SRTM-1: a finding that is consistent with other DEM assessments (Purinton and Bookhagen, 2017). The precision of the ASTER GDEM is particularly low compared to SRTM-1 (Czubski et al., 2013) and AW3D. However, this may depend on location as the ASTER GDEM quality is sensitive to a number of issues that vary spatially. This includes the number of stacked stereo-pair scenes for the DEM generation, as well as land and cloud cover, water masking and registration quality (Purinton and Bookhagen, 2017). In our study site in Nepal, the precision of river profiles derived from the TanDEM-X DEM is much lower than the reported 10 m absolute and 2 m relative height accuracy (Gruber et al., 2012a; Purinton and Bookhagen, 2017). Low accuracy along river profiles may relate to water bodies that are generally incoherent areas in the underlying DEM scenes and have height estimates that are known to be noisy and inaccurate (Wessel, 2016). The resolution of 12 m and high accuracy on hillslopes of the TanDEM-X DEM

(Purinton and Bookhagen, 2017) will offer new opportunities, but whether and how TanDEM-X DEMs will offer any advantage for the analysis of river profiles in high mountain areas compared to previous DEMs with lower resolution needs further study (Gruber et al., 2012b).

## 6.3 Error distribution and the CRS algorithm

The asymmetric distribution of residuals around the median (and mean) violate the assumptions of methods for filtering height errors that rely on least squares such as local regression techniques. Moreover, the large absolute values of outliers further complicate the application of methods that rely on averaging (e.g. running average filters). Although positive bias is on average more common, our results show that 10-50% of the profile values underestimate the actual river elevation. This underscores potential problems with using carving to hydrologically trim DEMs. Our new quantile-carving and CRS

algorithms implement a quantile regression that remains largely unaffected by skewed data and outliers, and thus prevents that very low or high values gain excessive weight for shaping the longitudinal profile.

Our comparison of river profiles derived from global DEMs with those derived from LiDAR shows that CRS-smoothing of river profiles can decrease differences to actual river elevations and gradients compared to commonly used methods of filling and carving. We derived optimal values of the smoothing parameter $K$ and the quantile $\tau$ by comparing the smoothed

profiles with LiDAR data that were available for subsets of the data and this may be a common approach where high-resolution, precise and accurate data is available for parts of the study area. Our results indicate, however, that optimal parameter values differ depending on whether the optimization's objective function minimizes errors between benchmark and smoothed elevations or gradients. Thus, the choice of objective function depends on specific uses and is crucial for calibrating parameters of the CRS algorithm.

Where benchmark data is unavailable the most suitable value of $K$ depends on the amount of scatter in the profiles, the true roughness (e.g., riffle-pool sequences in bedrock rivers vs. relatively smooth profiles of alluvial rivers), but also the type of application. One-dimensional hydraulic simulations, for example, may retain some of the variability of channel gradients while attempting to avoid the influence of erroneous data, and thus choose a low value of $K$. In many applications, visual interpretation of profiles will aid the identification of suitable values of $K$ and $\tau$. More general, it may be pragmatic to let $K$

be as large as possible while satisfying the constraints of the data (Sivia and Skilling, 2006). Although there exist ways to determine an optimal value of $K$ by cross-validation approaches (Garcia, 2010), qualitatively and visually cross-checking may be the most practical approach in many cases: too small values of $K$ will admit unnecessary and potentially erroneous structure whereas too large values impede a good agreement with the actual river profile (Sivia and Skilling, 2006).

## 6.4 Autocorrelated errors

River profiles are derived from measurements that give rise to errors of different types: random and systematic components, as well as artifacts (Reuter et al., 2009). We have shown that the CRS algorithm can efficiently handle random errors and may reduce offsets that arise from systematic or artefactual deviations between actual river profiles and those derived from

DEMs, thus improving an overall representation of longitudinal river elevations and gradients. Caution, however, must be taken with autocorrelated errors that we have not addressed here although they may significantly affect river profiles. Autocorrelation entails that if the true elevation at some location is overestimated in the DEM, then the elevation at a nearby pixel will likely also be overestimated (Temme et al., 2009). Autocorrelated errors have important consequences for the choice of smoothing parameters. In the case of short-range dependence, profiles may not be affected severely if a sufficiently large smoothness parameter is chosen. Long-range dependence, however, is neither easy to detect nor their structures simply removed by non-parametric regression. For example, it is difficult to ascribe the stepped patterns in Fig 3b to actual riffle-pool sequences or to artefacts that should be smoothed (Fig. 3c). Although approaches to nonparametric regression exist that are able to cope with autocorrelated errors (Opsomer et al., 2001), their implementation and assessment were, however, beyond the scope of this study. The CRS algorithm cannot differentiate actual steps in the profiles from artificial bumps. It smooths profiles to remove scatter and thus accentuates the actual patterns in the data. The quantile regression technique enables to derive uncertainty bounds which can support data interpretation by quantifying DEM errors.

### 6.5 Applications and future developments

The CRS algorithm can be used in various fields of research. Primarily, in studies of river profiles one may wish to remove noise from profiles while preserving underlying patterns and quantifying uncertainty. In tectonic geomorphology, for example, knickzones in longitudinal river profiles are essential proxies for transient river adjustment (Bishop et al., 2005), but distinguishing actual knickzones from data artifacts is challenging (Neely et al., 2017). Interquantile ranges determined by the CRS algorithm provide an objective way to determine minimum elevation drops that a knickzone must have to be identified against the background noise. Hydrodynamic simulations in mountainous environments frequently use globally available DEMs such as the SRTM or ASTER GDEM (Jarihani et al., 2015; Watson et al., 2015), but erroneous river profiles can introduce numerical instabilities (Paiva et al., 2011) and roughness, thus severely limiting the reliability of flood assessment, hazard zonation, and risk management (Watson et al., 2015). In a recent study on flash flood warning and hazard assessment in the Nepal Himalaya, a preliminary version of the CRS algorithm provided an important preprocessing step to improve the accuracy of estimating flow depth, flow speed, and flood wave arrival times (Bricker et al., 2017).

Applying quantile carving and the CRS algorithm is computationally more expensive compared to carving and filling. Run times depend largely on the number of nodes (or pixels) in a river network (Fig. 12). Thus, we applied our analysis to river networks only and not to entire DEMs, although this is principally possible provided convergent flow networks (O'Callaghan and Mark, 1984) have been derived. Rapid preprocessing of DEMs with our algorithms should apply parallelization strategies to allocate individual drainage basins or reaches to different processors. Future studies may also want to make use of the high flexibility of the algorithms. For example, values of the quantile $\tau$ and the smoothing parameter $K$ could be set for individual pixels or river reaches, thus varying the amount of smoothing spatially. This may be useful if the true variability in river gradients shall be retained where these are known to vary strongly, e.g., as a function of upslope area or hillslope relief (Fig. 9). Finally, the CRS algorithm is not restricted to elevation profiles, but can also be applied to

other variables measured or calculated along river profiles such as steepness (Ksn) or curvatures both of which are usually prone to even larger uncertainties.

## 7 Conclusions

Comparison of longitudinal river profiles derived from coarse-resolution globally available DEMs and LiDAR data shows that DEMs commonly overestimate elevations along rivers in steep topography. Moreover, error distributions become increasingly wider and positively skewed if adjacent hillslope gradients are steep. Our analysis suggests that the AW3D DEM has the highest precision and lowest bias for the analysis of river profiles in mountain topography. In contrast, the 12 m resolution TanDEM-X DEM has a very low precision, most likely due to the combined effect of steep valley-walls and the presence of water in valley bottoms. To characterize and remove such systematic and random errors in DEMs, we developed the CRS algorithm for correcting and smoothing longitudinal river profiles. The approach adopts a non-parametric quantile regression and handles entire river networks instead of single river reaches, and ensures downstream decreasing elevations. Compared to the conventional approaches of carving and filling, we find that our new approach is able to reduce the elevation bias and errors in longitudinal river profiles.

### Acknowledgments

We thank the German Aerospace Center (DLR) for granting access to TanDEM-X data as part of the project DEM_GEOL1053, and the OpenTopography Facility for providing a great platform for sharing LiDAR data and global DEMs. LiDAR data is based on services provided to the Plate Boundary Observatory by NCALM (http://www.ncalm.org). PBO is operated by UNAVCO for EarthScope (http://www.earthscope.org) and supported by the National Science Foundation (No. EAR-0350028, EAR-0732947, and EAR-1043051). We thank Fiona Clubb and an anonymous referee for their constructive comments.

## Appendix A:

### A1 River network representation, and along-river gradient and curvature

We represent the river network as a directed acyclic graph $G(V, E)$ (Heckmann et al., 2015). The set of graph nodes $V_i$ are pixel centers connected by directed edges $E(i, j)$. We denote the downstream node of $i$ as $j$ and the upstream node as $k$. While in general our algorithm is also applicable to divergent channel networks, i.e., braided or anastomosing rivers, as derived from multiple flow direction algorithms, we restrict our formulation to channel networks derived from the single flow direction algorithm (D8) (O'Callaghan and Mark, 1984). This restriction entails that $G$ is a directed tree network (Ahuja

et al., 1993) and each node $i$ has no more than one downstream neighbor $j$ and a maximum of eight upstream neighbors $k = \{1,2,\dots\}$ unless the node is the outlet or a channel head of the river network. We calculate the gradient of the river with forward finite differences

$$\frac{dz_i}{dx} = \frac{z_i - z_j}{x_i - x_j} \tag{A1}$$

so that positive gradients denote decreasing elevations in the downstream direction. In matrix notation, this equation is

$$\frac{d\mathbf{z}}{dx} = \mathbf{G}\mathbf{z} \tag{A2}$$

5    where elements on the main diagonal of $\mathbf{G}$ are $g_{ii} = (x_i - x_j)^{-1}$ and off-diagonal elements are $g_{ij} = -(x_i - x_j)^{-1}$.

We approximate the second derivative, i.e., the profile curvature, by the $2^{nd}$ order central difference.

$$\frac{d^2 z_i}{dx^2} = \frac{dz_k - dz_i}{0.5\,(x_k - x_j)dx} \tag{A3}$$

Since pixel centers may be cardinally and diagonally linked and are thus unequally spaced, we rearrange and write this equation in matrix notation for a reach with three nodes centered at $i$

$$\frac{d^2 z_i}{dx^2} = \left[\frac{2}{(x_i - x_j)(x_k - x_j)} \quad \frac{-2}{(x_k - x_i)(x_i - x_j)} \quad \frac{2}{(x_k - x_i)(x_k - x_j)}\right]\begin{bmatrix} z_j \\ z_i \\ z_k \end{bmatrix} \tag{A4}$$

or for all neighboring node triplets:

$$\frac{d^2 \mathbf{z}}{dx^2} = \mathbf{C}\mathbf{z} \tag{A5}$$

10    where each row $r$ in $C$ refers to an individual node triplet such that $C_{rj} = 2/((x_i - x_j)(x_k - x_j))$, $C_{ri} = -2/((x_k - x_i)(x_i - x_j))$, and $C_{rk} = 2/((x_k - x_i)(x_k - x_j))$.

**A2 Linear constrained regularized smoothing**

A linear least squares approach to longitudinal river profile smoothing can be derived as follows based on the variables
15    defined in Appendix A1. The approach uses a penalized least squares estimator that minimizes the sum of the squared residuals and its roughness at the same time. Given the roughness penalty $K>0$, the penalized sum of squares $PSS$ is

$$PSS(z) = \sum_{i=1}^{n} [z_i - \hat{z}_i]^2 + s \int_x [\hat{z}_i'']^2 dx \tag{A6}$$

The parameter $s$ determines the degree of exchange between the residual error and local variability of $\hat{z}$ and is calculated as

$$s = (\Delta x)^2 K \sqrt{\frac{n}{p}} \tag{A7}$$

from the smoothness parameter $K$, the spatial resolution $\Delta x$, the number of data points $n$ and the number of second derivative equations $p$. In matrix notation, Eq. (A6) is written as an overdetermined system of equations

$$\begin{bmatrix} I \\ sC \end{bmatrix} \hat{z} = \begin{bmatrix} z \\ 0 \end{bmatrix} \tag{A8}$$

or

$$A\hat{z} = b \tag{A9}$$

where $0$ is a vector of zeros with equal number of rows as $C$. We then derive the regularized elevations by the general linear model:

$$\hat{z} = (A^T A)^{-1} A^T b \tag{A10}$$

The resulting elevations will not necessarily decrease monotonically downstream. We thus reformulate the linear model in eq. (A10) as a quadratic programming problem that minimizes a linear least squares system under the constraint of monotonically decreasing elevations downstream:

$$\min_{\hat{z}} \frac{1}{2} \hat{z}^T H \hat{z} + f^T \hat{z} \text{ such that } \begin{cases} -G\hat{z} \leq -g_{min} \\ I_{eq} \cdot \hat{z} = z_{eq} \\ lb \leq \hat{z} \leq ub \end{cases} \tag{A11}$$

where the superscript $T$ refers to the transpose, $H = 2A^T A$ is a quadratic positive definite matrix and $f = -2A^T b$ represents the linear term. $I_{eq}$ is a $n$-by-$n$ matrix with ones on the main diagonal for nodes whose smoothed elevations must equal the values in $z_{eq}$. All other elements in $I_{eq}$ and $z_{eq}$ are zero. The linear constraint that $-G\hat{z} \leq -g_{min}$ ensures that elevation gradients at all nodes are equal or greater than $g_{min}$. Setting $g_{min} < 0$ will return longitudinal river profiles that are strictly monotonically downstream decreasing. The vectors $lb$ and $ub$ define lower and upper bounds of $\hat{z}$, respectively.

## A3 Quantile carving

Equation 1 can be efficiently solved by linear programming. Linear programming finds the vector $x$ that minimizes $f^T x$ subject to linear inequality ($Ax \leq b$) and equality constraints ($A_{eq} x = b_{eq}$) and lower bounds $lb$ (Mathworks, 2017):

$$\min_{\hat{z}} f^T x \text{ such that } \begin{cases} Ax \leq b \\ A_{eq} \cdot x = b_{eq} \\ lb \leq x \end{cases} \tag{A12}$$

If the stream network consists of $n$ nodes, $f = [(\tau\ \tau\ ...\ \tau)\quad 1 - (\tau\ \tau\ ...\ \tau)\quad (0\ 0\ ...\ 0)]^T$ with $n$ elements concatenated in each vector indicated by the round brackets. $Aeq = [I\quad -I\quad I]$ where $I$ is a $n \times n$ identity matrix and $beq = z$. $A = [0\quad 0\quad -G]$ with $0$ being a $n \times n$ matrix of zeros and $G$ is the gradient matrix introduced in Appendix A1, and $b = [(0\ 0\ ...\ 0)]^T$. The lower bounds are $lb = [(0\ 0\ ...\ 0)\quad (0\ 0\ ...\ 0)\quad -(\infty\ \infty\ ...\ \infty)]^T$. The linear program returns the vector $x$ whose elements $2n + 1$ to $3n$ refer to the elevation values $\hat{z}$.

## A4 CRS algorithm

Equation 3 is solved by quadratic programming. A quadratic programming problem has an objective which is a quadratic function of the variables in x:

$$\min_{\hat{z}} \frac{1}{2} x^T H x + f^T x \text{ such that } \begin{cases} Ax \leq b \\ A_{eq} \cdot x = b_{eq} \\ lb \leq x \end{cases} \tag{A13}$$

where the notation is the same as in Appendix A3. The quadratic term in the problem is

$$H = \begin{bmatrix} 0 & 0 & 0 \\ 0 & 0 & 0 \\ 0 & 0 & B \end{bmatrix} \tag{A14}$$

where $0$ being a $n \times n$ matrix of zeros and $B = 2(D^T D)$ and

$$D = \begin{bmatrix} 0 \\ sC \end{bmatrix} \tag{A15}$$

where $s$ determines the smoothness of the profile (Eq. A7) and $C$ is second order finite difference matrix introduced in Appendix A1. The quadratic program returns the vector $x$ whose elements $2n + 1$ to $3n$ refer to the elevation values $\hat{z}$.

## Appendix B

### Appendix B1: Software implementation and code availability

A software implementation of the CRS algorithm is available for the MATLAB software TopoToolbox on https://github.com/wschwanghart/topotoolbox. The software allows tuning several parameters. The parameter $K$ dictates the degree of smoothing. The choice of the value of $K$ involves a tradeoff between filtering river-profile variability present in reality and retaining data artifacts. Too high values of $K$ may introduce new artifacts. Estimating a suitable value of $K$ depends on both the type and magnitudes of errors in the data and the type of applications. Benchmark data such as LiDAR DEMs and precise (differential) global positioning system measurements can be used to estimate optimal parameter values that provide the best fit between the benchmark data and the DEM. If benchmark data is unavailable, visual crosschecking the results of the algorithm with variable parameters can guide the choice of suitable parameter settings. We developed a graphical user interface in TopoToolbox (*crsapp*) that supports a trial-and-error approach and visual crosschecking. Additional examples and applications of the algorithms can be found on http://topotoolbox.wordpress.com.

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

**Tables**

**Table 1: Digital elevation models (DEM) used in this study.**

|  | Yakima River, Central Washington | Big Tujunga, California | Kali Gandaki, Central Nepal |
|---|---|---|---|
| Benchmark DEM | Airborne LiDAR[1] | Airborne LiDAR[1] | - |
| DEMs | AW3D[3], ASTER GDEM[4], SRTM1[5] | AW3D[3], ASTER GDEM[4], SRTM1[5] | TanDEM-X DEM[2], AW3D[3], ASTER GDEM[4], SRTM1[5] |
| Coordinates | 46.83°N, 120.46°W | 34.17°N, 118.16°W | 28.16°N, 83.80°E |
| Topographic range (min, max) | 630 m (360 – 990 m a.s.l.) | 2000 m (300 – 2300 m a.s.l.) | 8050 m (50 – 8100 m a.s.l.) |

[1] Bare earth DEMs from the National Center for Airborne Laser Mapping; spatial resolution of 0.5 m; elevation accuracy 5-30 cm typical (±1-sigma); downloaded from http://www.opentopography.org

[2] Derived from X-band interferometric synthetic aperture radar data; spatial resolution of 12 m; absolute vertical accuracy <10 m (linear error at 90% confidence level); acquired, produced and provided by DLR (Wessel, 2016)

[3] ALOS World 3D 30; derived from ALOS PRISM stereo images; spatial resolution of 30 m; vertical error 5 m (root mean squared error); released by the Japan Aerospace Exploration Agency (JAXA) (Takaku et al., 2014) and downloaded from http://www.opentopography.org

[4] Version 2; spatial resolution of 30 m; ASTER GDEM is a product of NASA and METI; vertical accuracy 13 m (±1-sigma) (Tachikawa et al., 2011); downloaded from http://lpdaac.usgs.gov

[5] Shuttle Radar Topography Mission (Jarvis et al., 2008); spatial resolution of 30 m; absolute vertical accuracy <16 m (Falorni et al., 2005); downloaded from http://www.opentopography.org

**Table 2: Statistics of offsets between river-profile elevations derived from different global DEMs and benchmark LiDAR DEMs. $\tau_0$ refers to the quantile of the profile data where the offset between LiDAR and global DEMs is zero.**

|  |  | SRTM-1 | AW3D | ASTER GDEM |
|---|---|---|---|---|
| Yakima (low relief) | Mean (bias) [m] | 3.61 | 0.19 | 2.16 |
|  | Standard deviation [m] | 2.95 | 3.03 | 7.34 |
|  | Skewness | 0.46 | 3.95 | -0.61 |
|  | $\tau_0$ | 0.11 | 0.55 | 0.37 |
| Big Tujunga (medium relief) | Mean (bias) [m] | 15.93 | 5.90 | 10.08 |
|  | Standard deviation [m] | 8.11 | 4.98 | 8.23 |
|  | Skewness | 0.94 | 1.76 | 0.52 |

| | | $\tau_0$ | | 0.01 | 0.03 | 0.10 |
|---|---|---|---|---|---|---|

**Table 3: Statistical characterisation of residuals from smoothed river profiles derived with the CRS algorithm ($K$=5, $\tau$=0.5) from globally available DEMs and at different sites.**

| | DEM | Residual mean [m] | RMSE [m] | Minimum [m] | Maximum [m] | Skewness | Kurtosis |
|---|---|---|---|---|---|---|---|
| Yakima | SRTM-1 | -0.15 | 2.05 | -21.19 | 9.89 | -2.52 | 21.03 |
| (low relief) | AW3D | -0.17 | 1.78 | -19.00 | 7.77 | -3.61 | 32.96 |
| | ASTER GDEM | -0.23 | 4.51 | -37.00 | 17.12 | -1.28 | 11.85 |
| Big Tujunga | SRTM-1 | 0.05 | 3.40 | -27.60 | 23.59 | 0.14 | 12.28 |
| (medium relief) | AW3D | 0.16 | 2.97 | -34.44 | 38.83 | 0.90 | 29.03 |
| | ASTER GDEM | -0.06 | 3.89 | -29.85 | 28.93 | -0.20 | 8.54 |
| Kali Gandaki | SRTM-1 | 0.71 | 7.24 | -39.75 | 106.73 | 4.45 | 45.81 |
| (high relief) | AW3D | 0.05 | 1.97 | -12.20 | 40.98 | 4.96 | 72.61 |
| | ASTER GDEM | 0.48 | 15.20 | -78.99 | 114.22 | 0.56 | 8.86 |
| | TanDEM-X DEM | 1.19 | 18.79 | -236.84 | 172.84 | 2.36 | 27.41 |

15 **Table 4: Optimal values of $K$ and $\tau$ for CRS algorithm applied to the Yakima and Big Tujunga catchments and different DEMs. The root mean squared error (RMSE) is calculated from the deviations from the LiDAR data.**

| | DEM | K | $\tau$ | RMSE [m] | RMSE with CRS [m] | RMSE with carving [m] | RMSE with filling [m] |
|---|---|---|---|---|---|---|---|
| Yakima | SRTM-1 | 2.76 | 0.05 | 4.66 | 2.57 | 3.90 | 6.78 |
| (low relief) | AW3D | 0.53 | 0.40 | 3.04 | 1.48 | 5.61 | 14.40 |
| | ASTER GDEM | 21.93 | 0.37 | 7.65 | 3.86 | 12.71 | 13.73 |
| Big Tujunga | SRTM-1 | 1.57 | 0.01 | 17.88 | 11.68 | 16.16 | 20.33 |
| (medium relief) | AW3D | 1.00 | 0.06 | 7.72 | 5.41 | 7.01 | 10.01 |
| | ASTER GDEM | 2.57 | 0.03 | 13.01 | 7.92 | 11.77 | 14.37 |

**Figures**

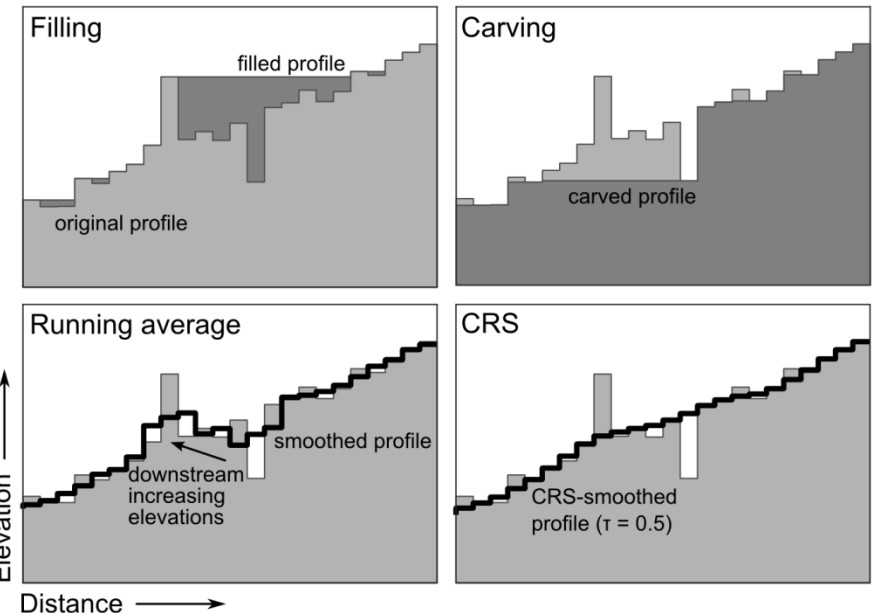

**Fig. 1: Detail of a longitudinal river profile and effects of different techniques to hydrologically correct and smooth the profile.**

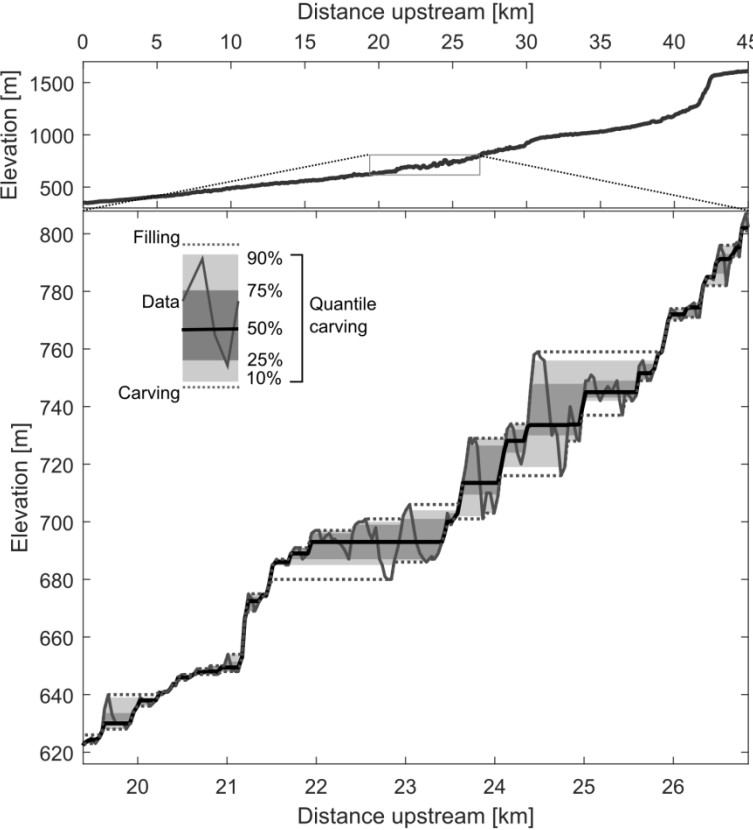

10 **Fig. 2: Quantile carving of the longitudinal profile of the Big Tujunga river. Quantile carving reconstructs the profile along different quantiles and thus continuously links the two common approaches carving and filling that run along minima and maxima of the data, respectively.**

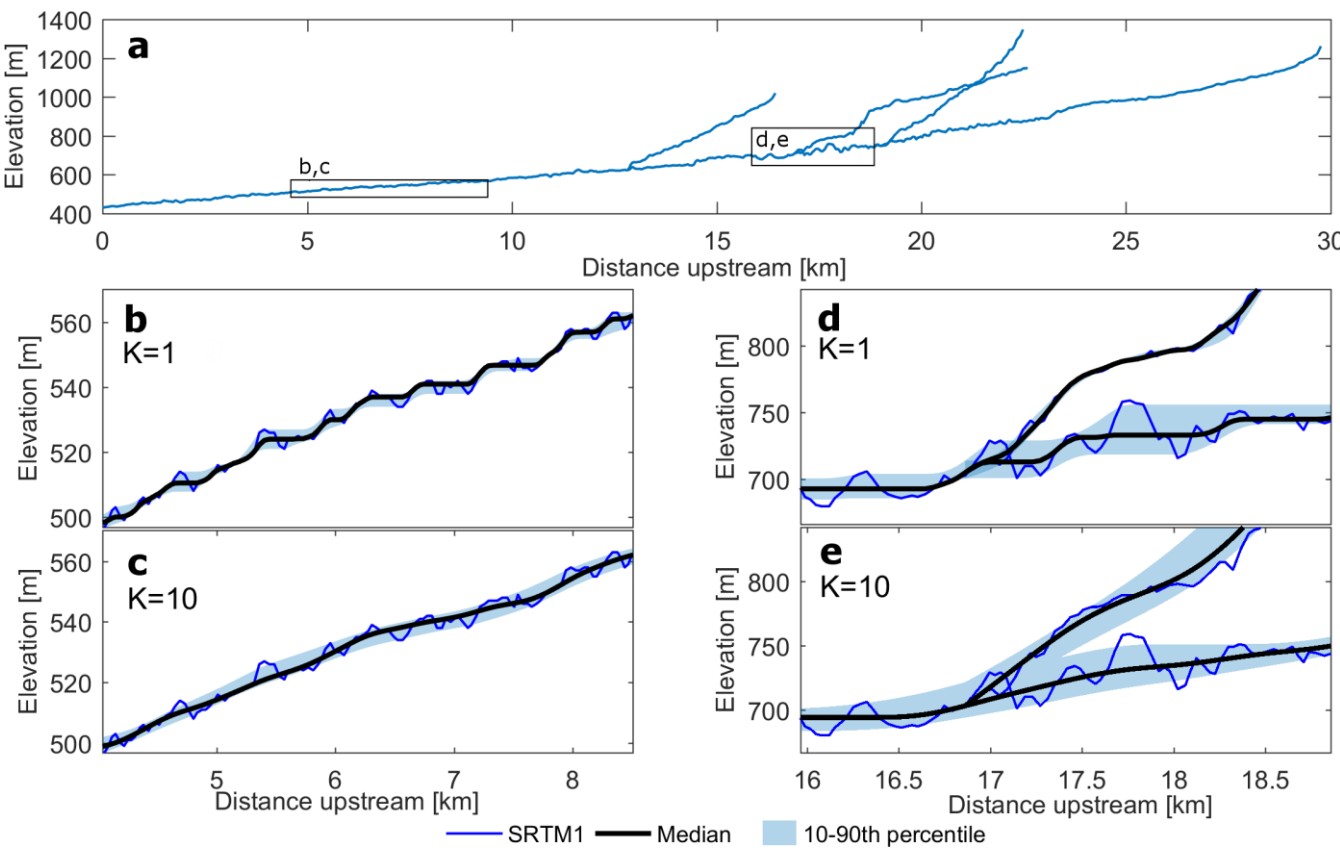

**Fig. 3: Applying the CRS-algorithm to selected streams in the Big Tujunga catchment (a). Panels b-e show details of the longitudinal river profile and results for the median and 10-90<sup>th</sup> percentile for the smoothing parameter *K*=1 and *K*=10.**

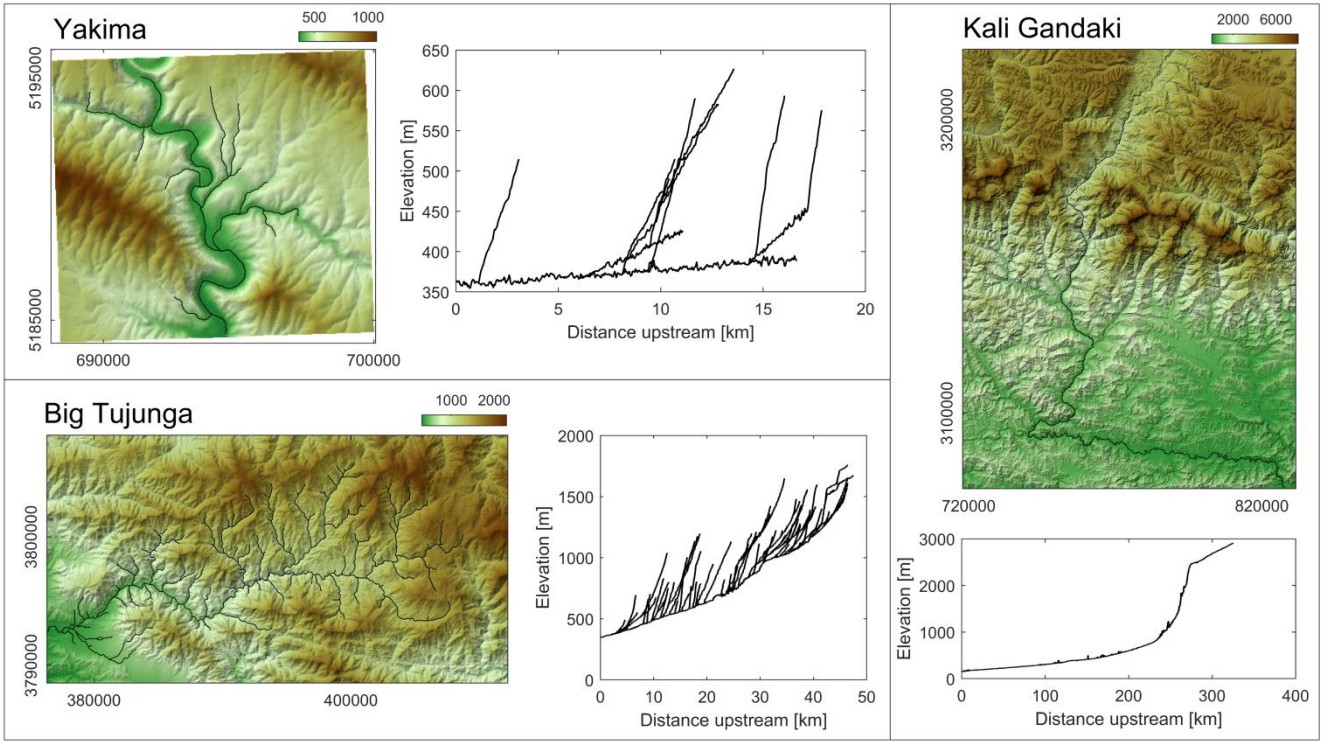

**Fig. 4: Sites and longitudinal river profiles in this study. Shaded reliefs are derived from SRTM-1 and colours indicate elevation in meters.**

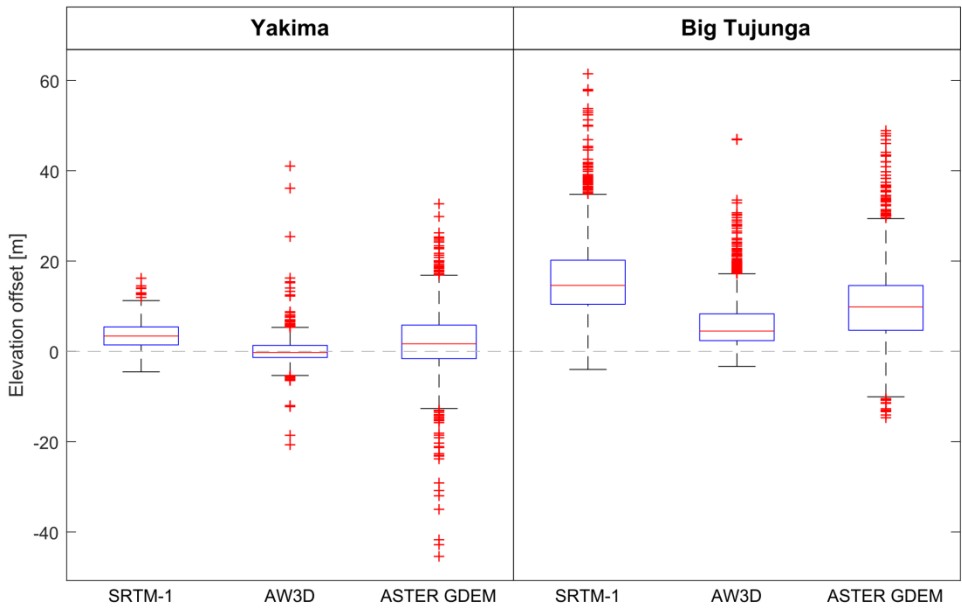

5    **Fig. 5: Elevation offsets between different DEMs and benchmark LiDAR DEMs calculated for all river pixels for the Yakima and Big Tujunga dataset (see Fig. 4).**

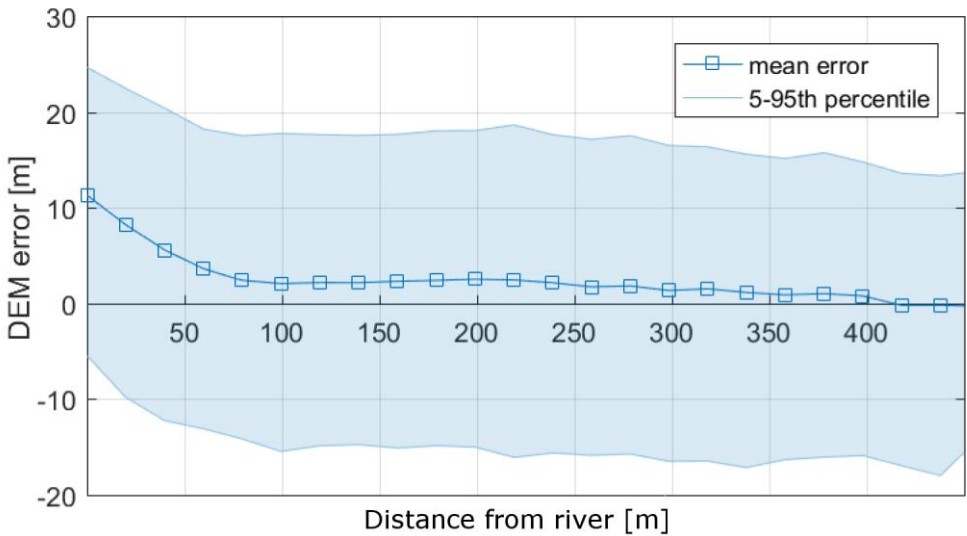

**Fig. 6: Bias of the SRTM-1 in the Big Tujunga catchment approaches zero with increasing distance from the river.**

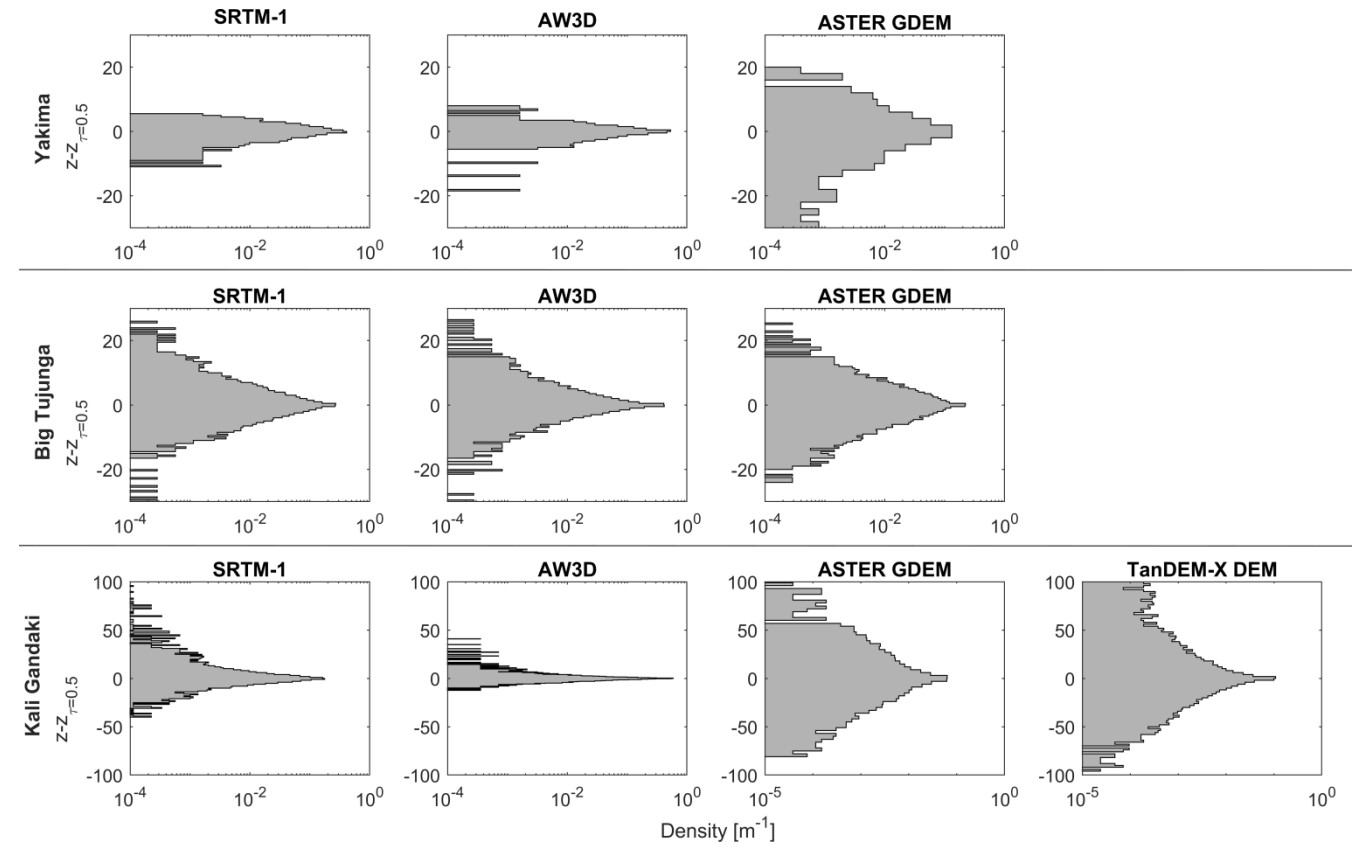

**Fig. 7:** Error distributions of longitudinal river profiles along the Yakima, Big Tujunga and Kali Gandaki calculated by the offset between measured elevations and elevations obtained with the CRS-algorithm with $\tau = 0.5$ and $K = 5$.

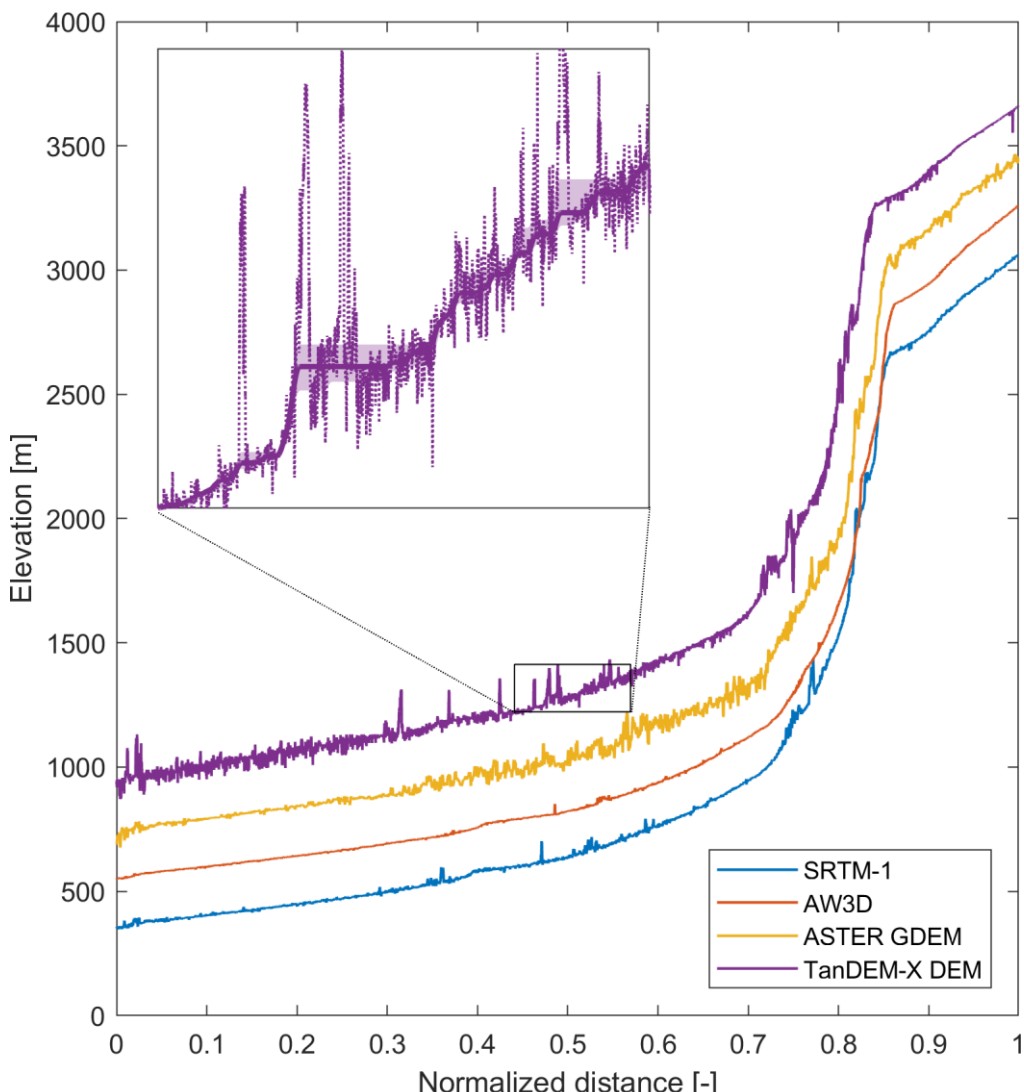

**Fig. 8: Profiles derived from different global DEMs for the Kali Gandaki, Nepal. To avoid overlap, we added an offset of 100 m between the different profiles. Horizontal distances along rivers vary and were thus normalized to range between 0 and 1. The inset shows details from the TanDEM-X DEM derived profile where the dashed line is the original data, the solid is the CRS-derived profile (K=10, τ = 0.5) and the shaded area delineates the interquartile range derived with the CRS algorithm (K=10, τ = 0.25 and 0.75).**

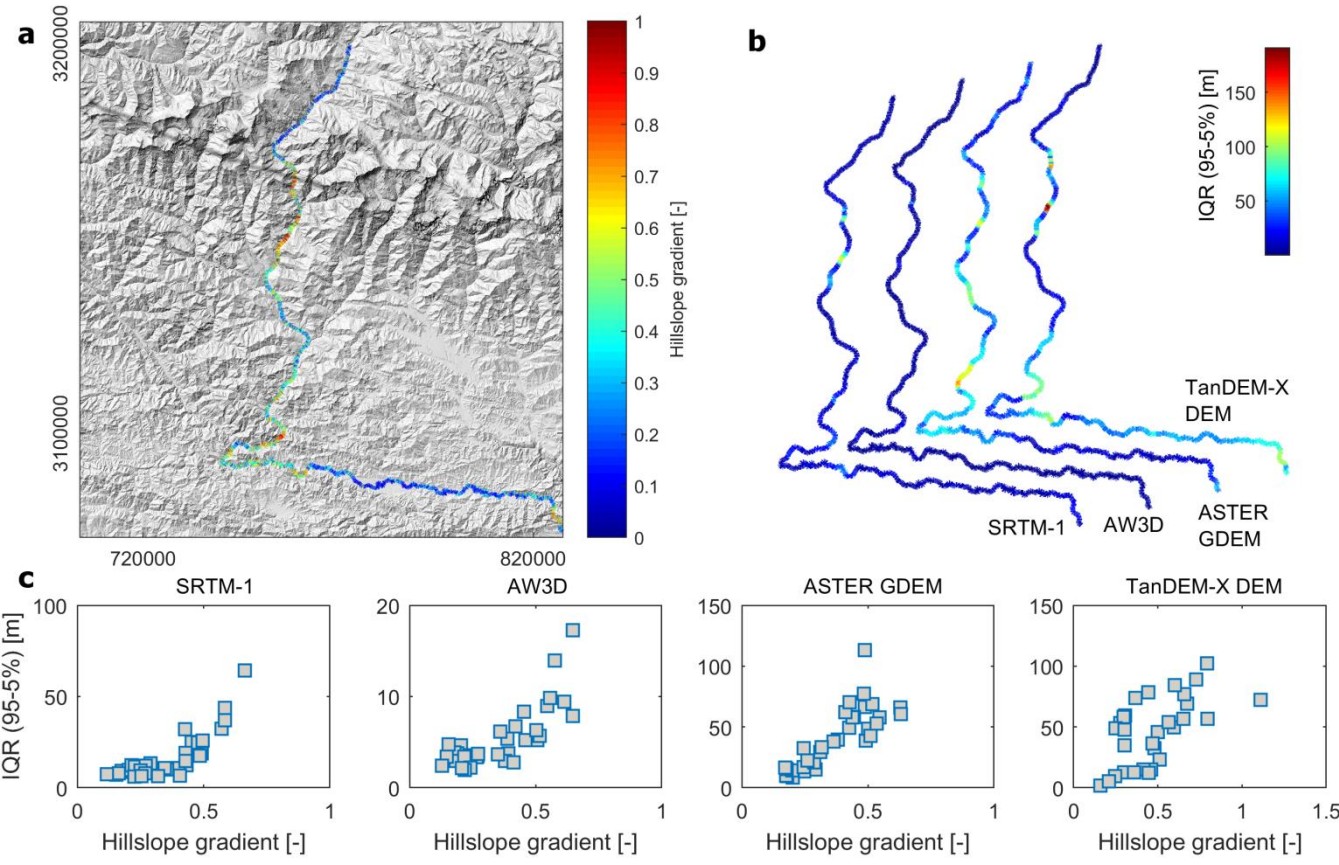

**Fig. 9: Planform patterns of DEM errors along the Kali Gandaki (a, b). The interquantile range (IQR) correlates with hillslope gradients adjacent to river (within <1000 m distance) (c).**

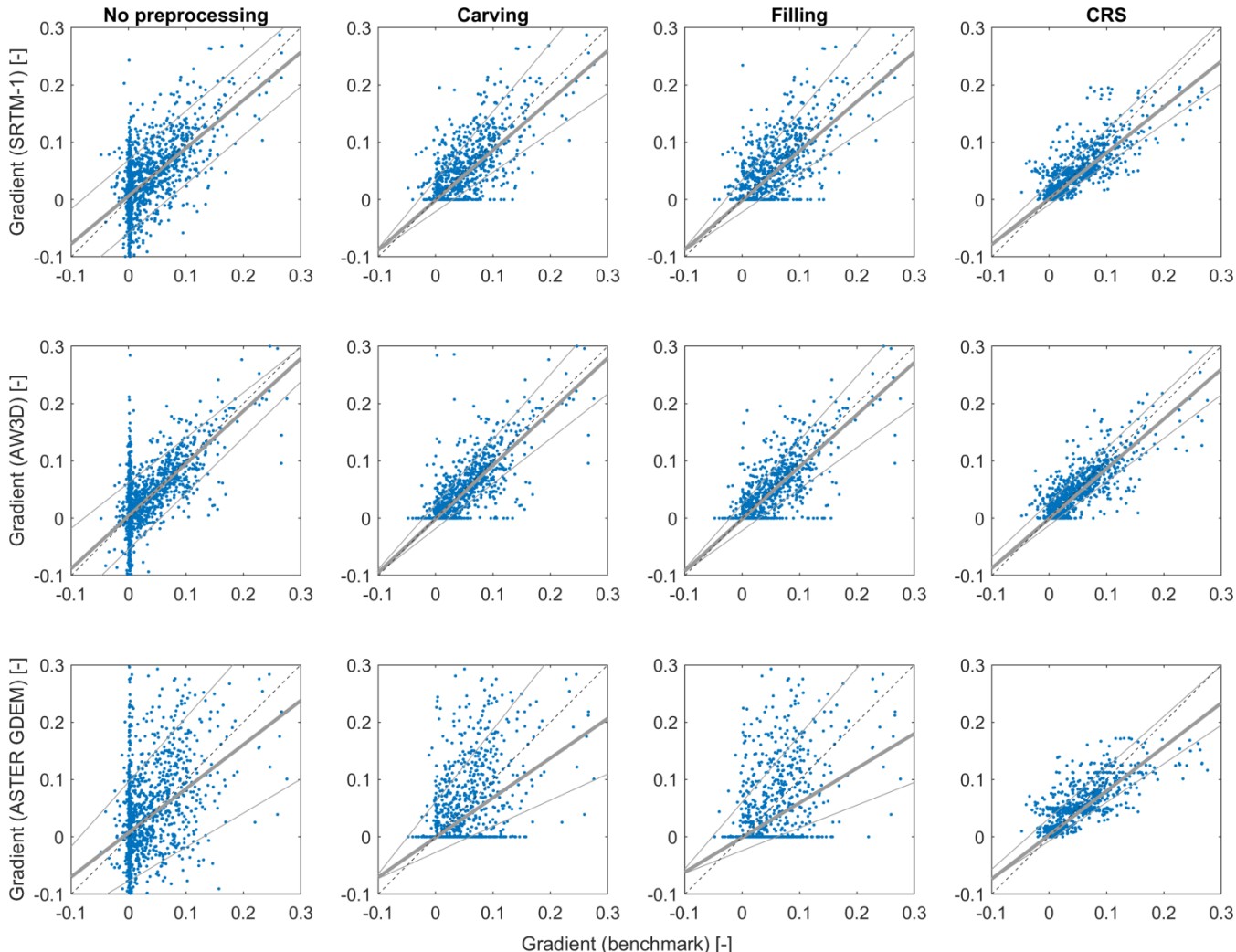

**Fig. 10:** Comparison of pixel-by-pixel along-river gradients derived from benchmark LiDAR DEMs (resampled to 30 m) and global DEMs in the Yakima catchment. Negative gradients in the benchmark and unprocessed DEMs are due to erroneously increasing river elevations in downstream direction. The grey solid lines (thin lines: 10 and 90%iles, thick line: median) are derived from a quantile regression. The dashed line is a one-to-one reference line. The parameters of the CRS algorithm (τ and K) are those in Table 4 of the manuscript. Compared to the original profiles and those obtained by preprocessing techniques of filling and carving, the CRS algorithm is able to reduce the differences between along-river gradients as indicated by the narrow interquantile (90-10%iles) range.

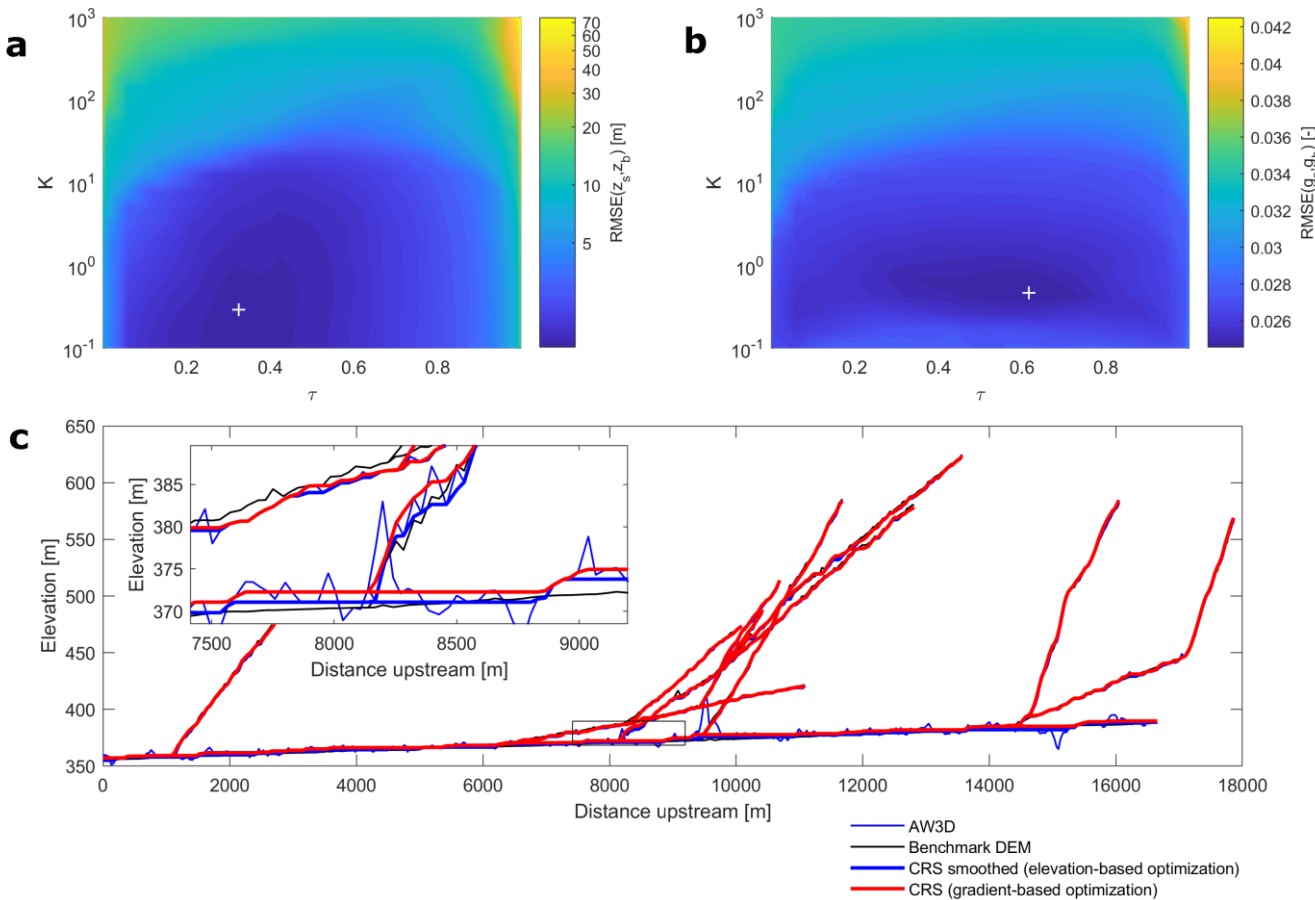

**Fig. 11: Sensitivity analysis of the CRS algorithm and the smoothing parameter K and quantile τ for the Yakima catchment and the AW3D DEM. a) Sensitivity of the root mean squared error (RMSE) between smoothed elevations and elevations from the LiDAR derived benchmark profile. A minimum RMSE is obtained for K = 0.26 and τ = 0.32 (white cross). b) Sensitivity of the RMSE between gradients derived from the smoothed profiles and the benchmark profile. A minimum RMSE is obtained for K = 0.43 and τ = 0.63 (white cross). c) Profile of the Yakima and its tributaries and detail (see inset).**

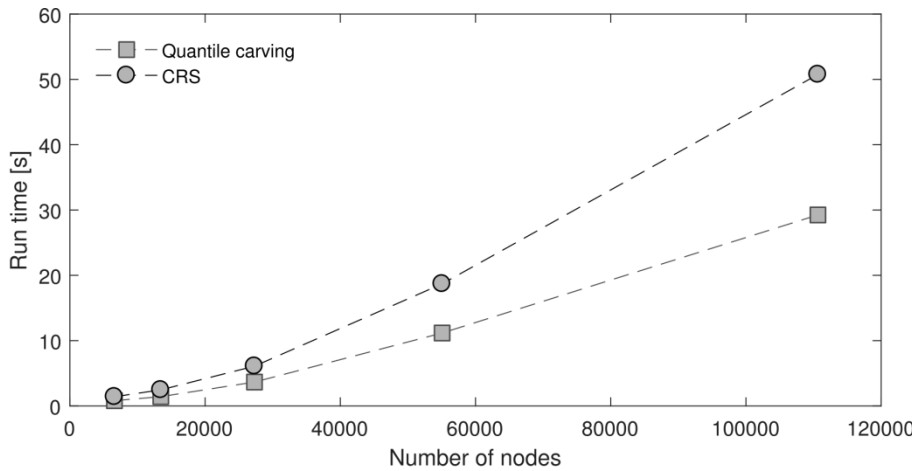

**Fig. 12: Run time of quantile carving and the CRS algorithm for river networks with different sizes. The number of nodes refers to the pixels in the DEM that constitute river pixels. Tests were run on a Windows 7 system with an Intel Xeon CPU (3.07 GHz, 4 cores) and 24 GB installed memory (RAM).**

