# Peer review of "Bumps in river profiles: uncertainty assessment and smoothing using quantile regression techniques"

_Earth Surface Dynamics, 2017_

## Referee Comment (RC1) · Anonymous Referee #1 · 30 Aug 2017

In this paper, the authors propose an innovative method to smoothen the longitudinal profiles derived from digital elevation models (DEMs) using a quantile-based statistics, called constrained regularized smoothing (CRS). The work is well presented, and the manuscript is ready to follow with an appropriate number of figures in good quality. The authors demonstrate extensive analysis on the proposed method using various kinds of global DEMs to find the pros and cons of the DEMs themselves, as well as to test the applicability of the smoothing method (CRS) to the noisy, DEM-derived channel profiles of both mainstream and tributaries. I believe this work is worth being published in the journal Earth Surface Dynamics, subject to some minor corrections.

.

.

[Figure]

{General comments}

**CRS-derived gradient:**

I think the paper would be strengthened if some additional demonstrations are provided. The authors show the nicely smoothed channel profiles, but do not provide the derivatives of the along-stream elevation (slope gradients). In the discussion, the authors state that "CRS-smoothing of river profiles can decrease differences to actual river elevations and gradients" (P10 L4), but the along-stream gradients after the CRS processing are not really provided. Furthermore, they mention that the CRS method will be useful for the analyses of knickzones and hydrolodynamics that often use stream gradient, and the representation of gradients can be highly of interest for many researchers working on fluvial (and other) processes. It can be more clearly demonstrated if they could show some examples from their own datasets – even just a visual representation (not a strict statistics) would help readers to understand the advantage of the CRS method in calculating the derivatives of elevation. I would, therefore, recommend adding a figure that shows not only the elevation profiles but also the slope (and curvature, if applicable) derived from the original and CRS-applied datasets.

**Title:**

The title sounds attractive, but not fully informative. Particularly, "the good, the bad, and the ugly" is vague. It would be better to include the key terms (such as global DEMs, quantile carving, and/or constrained regularized smoothing).

**Objectives:**

At the beginning of "4. Methods and data", the authors provide an explicit description of the goals of this study (P6 L22-L24), but these were not so clearly shown in the Introduction section (P2 L5-8). Please rewrite the objectives more clearly in Introduction.

.

.

{Specific comments}

P7 L4 "ellipsoidal heights (WGS84)" Which geoid was applied for each dataset (or the same for all)?

P7 L5 "resampled... resolution" To what resolution? How? (nearest neighbor?)

P10 L16 In this section the authors seem to discuss some sorts of errors. I do not figure out why they mention each error type as "good", "bad", and "ugly". I am sorry if I am missing some, but it would be better clarified – the differences of goodness, badness, and ugliness of each error.

P11 L6-7 Although I did not find any detailed descriptions of the smoothing method in Bricker et al. (2017), if the CRS algorithm (or a similar one) has already been presented in the previously published article, I think it should be explicitly shown prior to the methodological descriptions in this paper.

P11 L17 "other variables" such as...??

Table 4 It would be better to show that the values of RMSE are the deviations from the ALS data in this caption (not only in the main text).

Fig. 8 Please explain "Topographic shielding" in details. Does this corresponds to the "hillslope gradients adjacent to river within 1000 m distance"?

.

.

{Technical corrections}

P6 L13 The details of the place name appears later, but here please provide, at least, the region name "(San Gabriel Mountains, USA)" where "Big Tujunga catchment" locates.

P13 L11 "Eq. (11)" does not seem to appear elsewhere in the manuscript.

P14 L11-12 The numbering of the equation may be A12.

---

## Referee Comment (RC2) · F. Clubb (Referee) · 4 Sep 2017

**General comments**

This manuscript presents a new method for smoothing river long profiles based on a quantile-carving approach, which the authors then use to examine errors in elevation values extracted from different globally available DEMs. The paper is interesting and well-written, and provides useful insight into the applicability of different global DEMs for fluvial profile analysis as well as presenting a novel algorithm which has potential to be used in many studies. I think that the paper is suitable for publication in *Earth Surface Dynamics* following some minor corrections, which I have detailed below. Some general points:

[Figure]

- In many studies analysing river profiles, other metrics, such as channel gradient and drainage area, are used along with elevation to examine channel response to changes to external forcings such as climate or tectonics. It would be useful to include some analysis of how these metrics vary between the different DEM datasets, or with and without the CRS smoothing algorithm, as these are key datasets that will be needed in channel profile analysis by any users of the code. I suggest expanding the analysis (either by including a figure or another table) to include statistics of the channel gradients.

- The calibration of the parameters $K$ and $\tau$ clearly has a large impact on the elevation values extracted from the profiles (e.g. Fig 3). Although the authors discuss the fact that these parameters can affect the elevation values, it would be useful to include some more guidance on how these parameters can be set by the user to avoid over- or under-smoothing their channel profiles. This has been done for the $K$ parameter in Section 6.2, but it would be useful to also include information on the sensitivity of the method to $\tau$.

- I wonder if there is potential to include a spatially variable $K$ parameter along the profile based on the distribution of local relief (e.g. where the $K$ parameter is calculated directly from the relief of the surrounding landscape, rather than having to be set by the user)? This could be useful in areas like the Nepal site where the topography varies dramatically along profile. I'm not suggesting doing this for the paper, but it may be useful to include as a potential avenue of development for the method.

**Specific comments**

Page 1, Line 23: river profiles may also reflect signals of base level and sediment transport processes.

Page 2, Line 3: clarify here that bumps in river profiles may also be from real signals of climatic/tectonic perturbations in the profile.

Introduction: traditional methods of analysing river profiles to extract climatic/tectonic signals generally use slope and area as well as the elevation of the river profiles. It would be useful to mention these metrics in the introduction and how they relate to the extraction of the elevation values.

I think the introduction could be expanded to review some of the advantages of fluvial profile analysis and to include some more literature on how these methods have been applied in the past. At the moment I think this is slightly glossed over, and it would be good to emphasise this to show the value of the authors' new method for the geomorphology community.

Page 3, Line 32: I think it would be useful to expand upon these methods here and provide references for these approaches, since these are the benchmark algorithms which the authors are trying to improve upon with their method.

Page 4, Lines 2-5: Is this the running average approach demonstrated in Figure 1?

Page 4, Line 9: It would be helpful for the authors to restate the aims and approach of the study here to emphasise to the reader how the method that they outline improves upon previous methods given the research needs stated in this paragraph.

Figure 1: It would be good to highlight particular sections on the different profiles where the elevation increases downstream to make this clearer to the reader.

Page 4, Line 30: The authors state that the CRS method assumes local smoothness of the profile and spatial autocorrelation. I agree that a non-parameteric approach is very useful, as we don't have to assume that the channel is incision based on a specific model (e.g. stream power), but what is the justification for assuming that these should be local smoothness/spatial autocorrelation? We might expect this to be the case if the profile is in steady state and is nicely concave, but what about if the profile is transient? This will create local patches within the profile which are 'bumpy' or perhaps not spatially correlated. How easy is it for the method to differentiate between these and DEM artefacts?

Page 6, Eq 4: How does the sensitivity of the method vary with grid resolution ($\triangle x$)? From equation 4 it seems that the smoothing should increase as the resolution becomes coarser.

Fig 3: It looks like the smoothing parameter $K$ and $\tau$ have a big effect on the final shape of the profile. How should the user choose appropriate values of these parameters? This is maybe explained later in the paper.

Table 1: It would be useful to include the vertical errors on each of the datasets used in the study.

Page 7, Line 4 and throughout: Although the lidar DEMs are much higher resolution than the global datasets, it would still be useful to acknowledge/quantify the errors associated with these datasets. How were they gridded/filtered? What is the vertical error on the resulting DEMs? When was the lidar data flown compared to the global datasets (any temporal differences that could account for some of the error)?

Page 7, Lines 15-17: Would it be possible to vary the $K$ parameter spatially, for example, correlated with relief/gradient? This could be useful for sites such as the

Nepal one where there is a large difference in relief along the profile. May be beyond the scope of the paper for the moment, but it would be interesting to see if there was a correlation between the required smoothing parameter and the surrounding relief, so you could preferentially smooth the profile in areas more prone to errors.

Page 7, Lines 24-28 and Appendix B: I downloaded TopoToolbox and implemented the CRS smoothing method (which was easy to do!). It may be useful in the Appendix to include a link to a tutorial for using the CRS method, which I saw exists on the TopoToolbox wordpress. The authors have provided the tool *crsapp* for visual checking of the parameters, which is great, but it would be useful to provide a tutorial for use of this tool as well.

Figure 5: Could you expand upon this caption to show what is being represented here? Is this the offset for every pixel in the river profile?

Figure 7: It would be useful to see these data for the other two field sites as well as for the Nepal site, to show visually how the distribution of the residuals varies with relief.

Page 9, Section 6.2: I think including analysis of channel slope and curvature may be needed in order to compare the ability of the different DEM datasets to analyse topographic information. Although elevation may not vary much with grid resolution, parameters such as local slope and curvature have been shown to be very sensitive to grid resolution (e.g. Vaze et al., 2010 (Env. Modelling and Software); Grieve et al., 2016 (ESurf)), with the range of slope and curvature values decreasing with resolution. Analysis of elevation values alone may suggest that the TanDEM-X dataset is not an improvement on the 30m datasets, but the higher resolution dataset may actually be more useful for extraction of these other metrics which are also important for channel profile analysis.

Page 10, Section 6.3: In general I like the title, and I get the idea of the good, bad, and ugly errors, but I think it could be expanded on a bit more - the authors should clarify here why random errors are good, and systematic errors are bad (can state more clearly that random errors are easier to smooth from the profile, whereas systematic errors are more difficult to distinguish from real signals).

Page 10, Section 6.4: I think some of this section makes more sense to have in the introduction to set the context of why developing the CRS algorithm is important.

**Technical corrections**

Page 2, Lines 19-20: I'd suggest rewording this sentence - it's unclear at the moment. Do you mean that random errors may or may not be clustered spatially?

Page 6, Line 15: rephrase sentence 'Thus derived profiles are monotonously decreasing downstream while filtering the wiggles'

Page 9, Line 16-18: Split this into two sentences.

Appendix A3: Equation wrongly labelled, should be A12.

---

## Author Response (AR1)

**Earth Surface Dynamics Discussions (esurf-2017-50)**

**Bumps in river profiles: the good, the bad, and the ugly**
Wolfgang Schwanghart and Dirk Scherler

**Final response to comments by reviewers**

*Reply: We thank both reviewers for their helpful and constructive comments. Please find below our point-by-point replies to each of their comments. We have implemented the suggestions in the manuscript and show these changes as indented text together with our replies.*

**Reviewer 1:**

In this paper, the authors propose an innovative method to smoothen the longitudinal profiles derived from digital elevation models (DEMs) using a quantile-based statistics, called constrained regularized smoothing (CRS). The work is well presented, and the manuscript is ready to follow with an appropriate number of figures in good quality. The authors demonstrate extensive analysis on the proposed method using various kinds of global DEMs to find the pros and cons of the DEMs themselves, as well as to test the applicability of the smoothing method (CRS) to the noisy, DEM-derived channel profiles of both mainstream and tributaries. I believe this work is worth being published in the journal Earth Surface Dynamics, subject to some minor corrections.

*We thank Reviewer 1 for his/her constructive assessment of our work.*

{General comments}

**CRS-derived gradient:**

I think the paper would be strengthened if some additional demonstrations are provided. The authors show the nicely smoothed channel profiles, but do not provide the derivatives of the along-stream elevation (slope gradients). In the discussion, the authors state that "CRS-smoothing of river profiles can decrease differences to actual river elevations and gradients" (P10 L4), but the along-stream gradients after the CRS processing are not really provided. Furthermore, they mention that the CRS method will be useful for the analyses of knickzones and hydrolodynamics that often use stream gradient, and the representation of gradients can be highly of interest for many researchers working on fluvial (and other) processes. It can be more clearly demonstrated if they could show some examples from their own datasets – even just a visual representation (not a strict statistics) would help readers to understand the advantage of the CRS method in calculating the derivatives of elevation. I would, therefore, recommend adding a figure that shows not only the elevation profiles but also the slope (and curvature, if applicable) derived from the original and CRS-applied datasets.

*We thank Reviewer 1 for this suggestion. In the revised manuscript, we provide additional visualizations that allow readers to interpret the results from the CRS algorithm. Derivatives of longitudinal river profiles (e.g. along-river gradients) were also addressed by Reviewer 2 and we detail some findings below that we will include in the revised manuscript.*

**Title:**

The title sounds attractive, but not fully informative. Particularly, "the good, the bad, and the ugly" is vague. It would be better to include the key terms (such as global DEMs, quantile carving, and/or constrained regularized smoothing).

*We changed the title to: Bumps in river profiles: uncertainty assessment and smoothing using quantile regression techniques*

**Objectives:**

At the beginning of "4. Methods and data", the authors provide an explicit description of the goals of this study (P6 L22-L24), but these were not so clearly shown in the Introduction section (P2 L5-8). Please rewrite the objectives more clearly in Introduction.

*We have rewritten the aims and objectives in the introduction:*

> *The objective of this study is to characterize and quantify the uncertainties of elevation values in longitudinal river profiles derived from near-globally and publicly available DEMs including the new TanDEM-X DEM. To attain this goal, we devise new algorithms (quantile carving and constrained regularized smoothing) that use non-parametric quantile regression for assessing uncertainties and smoothing of river profiles. Using LiDAR DEMs as benchmark data and our new algorithms, we study how longitudinal river profiles derived from globally available DEMs (Table 1) are affected by errors and how these errors depend on the topographic setting. Moreover, we examine the best choice of parameter values for our algorithms to guide their application. Our algorithms will aid the visual interpretation and automated analysis of longitudinal river profiles and has additional applications in hydrodynamic modelling.*

{Specific comments}

P7 L4 "ellipsoidal heights (WGS84)" Which geoid was applied for each dataset (or the same for all)?

*Please see subsequent comment below.*

P7 L5 "resampled... resolution" To what resolution? How? (nearest neighbor?)

*We rewrote this sentence to clarify the geodetic height datum and the type of resolution:*

> *"We used the reference ellipsoid defined by the world geodetic system WGS 84 as basis for all DEMs and resampled them to the same spatial extent and 30 m resolution using bilinear interpolation".*

P10 L16 In this section the authors seem to discuss some sorts of errors. I do not figure out why they mention each error type as "good", "bad", and "ugly". I am sorry if I am missing some, but it would be better clarified – the differences of goodness, badness, and ugliness of each error.

*We rewrote this section and avoided the terms good, bad and ugly.*

*River profiles are derived from measurements that give rise to errors of different types: random and systematic components, as well as artifacts (Reuter et al., 2009). We have shown that the CRS algorithm can efficiently handle random errors and may reduce offsets that arise from systematic or artefactual deviations between actual river profiles and those derived from DEMs, thus improving an overall representation of longitudinal river elevations and gradients. Caution, however, must be taken with autocorrelated errors that we have not addressed here although they may significantly affect river profiles. Autocorrelation entails that if the true elevation at some location is overestimated in the DEM, then the elevation at a nearby pixel will likely also be overestimated (Temme et al., 2009). Autocorrelated errors have important consequences for the choice of smoothing parameters. In the case of short-range dependence, profiles may not be affected severely if a sufficiently large smoothness parameter is chosen. Long-range dependence, however, is neither easy to detect nor their structures simply removed by non-parametric regression. For example, it is difficult to ascribe the stepped patterns in Fig 3b to actual riffle-pool sequences or to artefacts that should be smoothed (Fig. 3c). Although approaches to nonparametric regression exist that are able to cope with autocorrelated errors (Opsomer et al., 2001), their implementation and assessment were, however, beyond the scope of this study.*

P11 L6-7 Although I did not find any detailed descriptions of the smoothing method in Bricker et al. (2017), if the CRS algorithm (or a similar one) has already been presented in the previously published article, I think it should be explicitly shown prior to the methodological descriptions in this paper.

*The study of Bricker et al. (2017) used a previous version of the algorithm and was applied to a single reach and not an entire river network and not tested in detail. We thus refrained from a detailed mathematical description of the algorithm in that paper. Nevertheless, we changed the wording in this sentence:*

*In a recent study on flash flood warning and hazard assessment in the Nepal Himalaya, a preliminary version of the CRS algorithm provided an important preprocessing step to improve the accuracy of estimating flow depth, flow speed, and flood wave arrival times (Bricker et al. 2017).*

P11 L17 "other variables" such as...??

*We added examples as requested:*

*Finally, the CRS algorithm is not restricted to elevation profiles, but can also be applied to other variables measured or calculated along river profiles such as steepness (Ksn) or curvatures both of which are usually prone to even larger uncertainties.*

Table 4 It would be better to show that the values of RMSE are the deviations from the ALS data in this caption (not only in the main text).

*Yes, we have added this information to the caption of Table 4:*

*The root mean squared error (RMSE) is calculated from the deviations from the LiDAR data.*

Fig. 8 Please explain "Topographic shielding" in details. Does this corresponds to the "hillslope gradients adjacent to river within 1000 m distance"?

*According to Codilean (2006), topographic shielding refers to the proportion of the incoming cosmic radiation that is shielded by the surrounding topography. It is used in cosmogenic nuclide dating to correct for decreased production rates in steep terrain. We accidently plotted topographic shielding instead of hillslope gradient in Fig. 8 of the manuscript, and we have now changed it to hillslope gradient within 1000 m distance from the stream network. The patterns, however, are similar as both measures are highly correlated and thus the changes do not affect our conclusions.*

{Technical corrections}

P6 L13 The details of the place name appears later, but here please provide, at least, the region name "(San Gabriel Mountains, USA)" where "Big Tujunga catchment" locates.

*Done*

P13 L11 "Eq. (11)" does not seem to appear elsewhere in the manuscript.

*Eq 11 has been changed to Eq A6.*

P14 L11-12 The numbering of the equation may be A12.

*Done*

*References:*

*Codilean, A.T. (2006): Calculation of the cosmogenic nuclide production topgrahic shielding scaling factor for large areas using DEMs. Earth Surface Processes and Landforms, 31, 785-794.*

**Reviewer 2 (Fiona Clubb):**

This manuscript presents a new method for smoothing river long profiles based on a quantile-carving approach, which the authors then use to examine errors in elevation values extracted from different globally available DEMs. The paper is interesting and well-written, and provides useful insight into the applicability of different global DEMs for fluvial profile analysis as well as presenting a novel algorithm which has potential to be used in many studies. I think that the paper is suitable for publication in Earth Surface Dynamics following some minor corrections, which I have detailed below. Some general points:

- In many studies analysing river profiles, other metrics, such as channel gradient and drainage area, are used along with elevation to examine channel response to changes to external forcings such as climate or tectonics. It would be useful to include some analysis of how these metrics vary between the different DEM datasets, or with and without the CRS smoothing algorithm, as these are key datasets that will be needed in channel profile analysis by any users of the code. I suggest expanding the analysis (either by including a figure or another table) to include statistics of the channel gradients.
  *We agree with Reviewer 2 that addressing along-river gradients more explicitly would be*

*useful. We have done additional analysis on gradients that we detail further below in this reply and that are included in the revised version of the manuscript.*

- The calibration of the parameters K and τ clearly has a large impact on the elevation values extracted from the profiles (e.g. Fig 3). Although the authors discuss the fact that these parameters can affect the elevation values, it would be useful to include some more guidance on how these parameters can be set by the user to avoid over- or under-smoothing their channel profiles. This has been done for the K parameter in Section 6.2, but it would be useful to also include information on the sensitivity of the method to τ.
  *Profiles for different values of tau are shown in Fig. 3. Yet, the reviewer is right that we have not explicitly addressed the sensitivity of the smoothed profiles to changes in tau, largely because tau does not primarily affect the form of the profile but rather vertically shifts the profile. However, for very large (>0.99) or low (<0.01) values of tau, the algorithm shows some misfits to the data in particular if there are only few data points in the profile. We interpret this behavior to be related to difficulties of deriving quantiles for a limited amount of data points (see our sensitivity test below). We have addressed this issue in the revised version of the manuscript and included a sensitivity analysis.*

- I wonder if there is potential to include a spatially variable K parameter along the profile based on the distribution of local relief (e.g. where the K parameter is calculated directly from the relief of the surrounding landscape, rather than having to be set by the user)? This could be useful in areas like the Nepal site where the topography varies dramatically along profile. I'm not suggesting doing this for the paper, but it may be useful to include as a potential avenue of development for the method.
  *We thank Reviewer 2 for this suggestion. In fact, we mention this possibility in the last paragraph of the discussion. We do not plan to include spatially variable K or tau in a revised version. This would require deriving a statistical model between e.g. hillslope gradients and K which is beyond the scope of this paper. However, we consider implementing such option in a newer version of TopoToolbox in the future.*

Specific comments

Page 1, Line 23: river profiles may also reflect signals of base level and sediment transport processes.

*We have changed the sentence accordingly:*

> *The geometry of a river and specifically its longitudinal profile, reflect the climatic and tectonic forcing, variations in base level and sediment transport processes as well as differences in bedrock erodibility.*

Page 2, Line 3: clarify here that bumps in river profiles may also be from real signals of climatic/tectonic perturbations in the profile.

*We have changed the second half of the paragraph according to the suggested change:*

> *Structures such as bridges, culverts and reservoirs affect longitudinal river profiles derived from DEMs in ways that can either hide features present in reality or introduce patterns*

*that do not represent the actual course of the profile (Schwanghart et al., 2013). Deviations from graded river profiles may reflect signals of climatic or tectonic perturbations, but often they relate to errors and artifacts that generate bumpy river profiles and thus introduce uncertainties that may compromise the interpretation of longitudinal river profiles (Hayakawa and Oguchi, 2006; Wobus et al., 2006).*

Introduction: traditional methods of analysing river profiles to extract climatic/tectonic signals generally use slope and area as well as the elevation of the river profiles. It would be useful to mention these metrics in the introduction and how they relate to the extraction of the elevation values.

**Yes, we mentioned these metrics in the revised the manuscript (see next comment).**

I think the introduction could be expanded to review some of the advantages of fluvial profile analysis and to include some more literature on how these methods have been applied in the past. At the moment I think this is slightly glossed over, and it would be good to emphasise this to show the value of the authors' new method for the geomorphology community.

**We have tried to keep the introduction as short as possible and instead chose for a more detailed account of DEM errors and their effects on river profile analysis in the second section of the manuscript. However, in order to make the paper more accessible to other readers, we expanded our discussion of the literature on fluvial profile analysis in the first paragraph of the introduction.**

> *Rivers play a dominant role in the topographic evolution of the Earth surface, and possibly other planetary bodies (Hack, 1957; Howard, 1998; Whipple et al., 2013). They transfer sediment from mountains to depositional basins, set the base level for hillslopes, and convey tectonic and climatic signals across landscapes. The geometry of a river and specifically its longitudinal profile, reflect the climatic and tectonic forcing, variations in base level and sediment transport processes as well as differences in bedrock erodibility. Gradients along rivers, for example, reflect spatial variations in uplift rates (Whipple et al., 2013; Mudd et al., 2014; Scherler et al., 2014) and indicate the extent of past glaciations (Brardinoni and Hassan, 2006). Moreover, they can act as predictors for the zones of erosion and sediment accumulation during extreme events (Devrani et al., 2015) and reflect the repeated impact of masswasting events (Korup, 2006). Longitudinal river profiles and metrics derived them (e.g., the normalized channel steepness metric $k_{sn}$ (Wobus et al., 2006)) have become important tools for studying the topographic evolution of mountain belts and deciphering changes in climate and tectonics (Bishop et al., 2005).*

Page 3, Line 32: I think it would be useful to expand upon these methods here and provide references for these approaches, since these are the benchmark algorithms which the authors are trying to improve upon with their method.

**We added several references to these methods.**

Page 4, Lines 2-5: Is this the running average approach demonstrated in Figure 1?

**No, Fig. 1 shows a simple running average approach whereas Aiken and Brierley (2013) used a robust version of local regression using weighted linear least squares and a $1^{st}$-degree polynomial**

*model. Our reference to Fig. 1 should illustrate that profiles may contain sections with downstream increasing elevations. We realized that this is ambiguous. We thus rewrote these sentences:*

> *Finally, even robust approaches that aim at reducing the influence of individual outliers on the smoothed curve (Aiken and Brierley, 2013) may yield profiles that contain sections with downstream increasing elevation that result from long sections where valley bottom elevations are overestimated. An example of how outliers may generate downstream increasing elevations in profiles smoothed by a running average is shown in Fig. 1c.*

Page 4, Line 9: It would be helpful for the authors to restate the aims and approach of the study here to emphasise to the reader how the method that they outline improves upon previous methods given the research needs stated in this paragraph.

*We agree that it is a good idea to restate the aims after having defined the problems that we derive in the state-of-the-art section. In the revised version of the manuscript, we restate the aims of the study at the beginning of the methods section.*

Figure 1: It would be good to highlight particular sections on the different profiles where the elevation increases downstream to make this clearer to the reader.

*We changed Fig. 1 following this comment.*

Page 4, Line 30: The authors state that the CRS method assumes local smoothness of the profile and spatial autocorrelation. I agree that a non-parameteric approach is very useful, as we don't have to assume that the channel is incision based on a specific model (e.g. stream power), but what is the justification for assuming that these should be local smoothness/spatial autocorrelation? We might expect this to be the case if the profile is in steady state and is nicely concave, but what about if the profile is transient? This will create local patches within the profile which are 'bumpy' or perhaps not spatially correlated. How easy is it for the method to differentiate between these and DEM artefacts?

*Good point! The justification for assuming spatial autocorrelation or local smoothness is the First Law of Geography (also termed Tobler's law), which states that "near things are more related than distant things", an assumption that underlies all interpolation and smoothing techniques. Spatial autocorrelation is indeed vital for any kind of spatial analysis (De Smith et al., 2007) and should guide the value of a smoothing parameter together with the question and spatial scale underlying an analysis. In river profile analysis, our primary assumption is that of downstream decreasing elevations which justifies flood filling, carving and quantile carving. Local smoothness provides an additional assumption that is supported by the notion that river profiles usually do not exhibit the stepped profiles that appear in DEM-derived profiles, a notion that is supported by the analysis of better and higher resolution data in this study. Yet, Reviewer 2 is right in that she argues that transient river profiles may contain bumps or lack spatial autocorrelation. In fact, river profiles may have riffle-pool sequences or stationary knickpoints even if they are in steady state. Can CRS differentiate these real bumps from artefacts? No. CRS is not a classification algorithm. It smooths the profiles to remove scatter while accentuating the actual patterns in the data. However, quantile regression enables to derive uncertainty bounds which will support data interpretation and quantify DEM artefacts.*

Page 6, Eq 4: How does the sensitivity of the method vary with grid resolution (4x)? From equation 4 it seems that the smoothing should increase as the resolution becomes coarser.

*Good point. In fact, the squared spatial resolution (Δx) in Eq. 4 renders the degree of smoothing independent of the spatial resolution as it cancels out the squared distances in Eq. A4. Eq. 4 thus entails that a profile smoothed by the algorithm with a smoothing parameter K is insensitive to a linear transformation of the distance x.*

*Conversely, if we resample a given profile to a higher resolution, then a different value of K is required to obtain the same or at least similar smoothing results to those of the smoothed original profile. Here, the values of s (see Eq. 4) must be the same. For two profiles (A and B) with different spatial resolution $\Delta x_A$ and $\Delta x_B$, and smoothing parameter $K_A$, the smoothing parameter $K_B$ provides similar or same profiles if calculated by*

$$K_B = (\Delta x_A / \Delta x_B)^2 \, K_A$$

*We tested this relation using the following MATLAB script*

```
%% Load data
DEM = GRIDobj('srtm_bigtujunga30m_utm11.tif');
FD = FLOWobj(DEM);
% Stream network A
SA  = STREAMobj(FD,'minarea',500000,'unit','map');
SA  = klargestconncomps(trunk(SA));
% Stream network B
DEMB = resample(DEM,15);
FD  = FLOWobj(DEMB);
SB  = STREAMobj(FD,'minarea',500000,'unit','map');
SB  = klargestconncomps(trunk(SB));

% Smooth
K_A = 5;
zsA  = crs(SA,DEM,'K',K_A,'split',false);
zsB = crs(SB,DEMB,'K',(DEM.cellsize./DEMB.cellsize)^2*K_A,'split',false);

% Plot
plotdz(SA,DEM,'color','b')
hold on
plotdz(SB,DEMB,'color','k')
plotdz(SA,zsA,'color','b','LineWidth',2)
plotdz(SB,zsB,'color','k','LineWidth',2)
hold off

legend('original profile A (30 m resolution)',...
       'profile B from resampled DEM (15 m)', ....
       'smoothed profile (30 m, K_A = 5)', ...
       'smoothed profile (15 m, K_B = (30/15)^2 K_A')
```

[Figure]

*Fig. 1: Comparison of CRS-smoothed profiles of the Big Tujunga river derived from the original SRTM-1 (30 m resolution, A) and a resampled DEM (15 m, B). To obtain a similar smoothing result for B, $K_B$ must be adjusted to take account for the differences in spatial resolution. The slight horizontal offset between the profiles is an artefact that arises from changed flow distances due to DEM resampling.*

**We added the following text together with the above Equation (as Eq. 5):**

> **"Larger values of the smoothing parameter K result in smoother profiles (Fig. 3). We include the squared spatial resolution into the equation so that any linear transformation (e.g. distance scaling) of the river profile does not affect the smoothing results. Yet, this entails that profiles with different spatial resolutions must be smoothed with different values of K to obtain the same or at least similar results. If two profiles A and B have different spatial resolutions $\Delta x\_A$ and $\Delta x\_B$, then smoothing both profiles will return similar results if the smoothing parameter $K\_B$ is calculated from $K\_A$ by Eq. (5)."**

Fig 3: It looks like the smoothing parameter K and τ have a big effect on the final shape of the profile. How should the user choose appropriate values of these parameters? This is maybe explained later in the paper.

**We have added a sensitivity analysis that illustrates the effect of variable K and tau on both elevation and gradients obtained from longitudinal river profiles. We added following figure to the manuscript to illustrate the sensitivity analysis and describe them in the methods, results and discussion.**

[Figure]

*Fig. 2: Sensitivity analysis of the CRS algorithm and the smoothing parameter K and quantile tau for the Yakima catchment and the AW3D DEM. a) Sensitivity of the root mean squared error (RMSE) between smoothed elevations and elevations from the LiDAR derived benchmark profile. b) Sensitivity of the RMSE between gradients derived from the smoothed profiles and the benchmark profile. c) Profile of the Yakima and its tributaries and detail (see inset).*

Table 1: It would be useful to include the vertical errors on each of the datasets used in the study.

***We added the vertical errors to the footnotes in Table 1.***

Page 7, Line 4 and throughout: Although the lidar DEMs are much higher resolution than the global datasets, it would still be useful to acknowledge/quantify the errors associated with these datasets. How were they gridded/filtered? What is the vertical error on the resulting DEMs? When was the lidar data flown compared to the global datasets (any temporal differences that could account for some of the error)?

***The LiDAR DEMs used in this study have an elevation accuracy of 5-30 cm, ±1-sigma. The DEMs are bare-earth DEMs and their derivation using the software TerraScan is explained in the processing reports available on opentopography.org. The Yakima data was acquired in 2008 and the Big Tujunga data obtained in 2007, which is later than the acquisition of the SRTM and overlaps or is prior to acquisition of the ALOS and ASTER data. We cannot exclude any temporal differences although we expect that these were minor and local. We added the following text:***

> ***The LiDAR DEMs are bare-earth DEMs derived from point clouds (>3 points/m$^2$) and were downloaded from the OpenTopography facility (Table 1). Due to their decimeter to sub-decimeter accuracy, we consider the LiDAR data as our benchmark DEMs. The acquisition dates of the DEMs vary but we expect that temporal changes in the land surface are minor.***

Page 7, Lines 15-17: Would it be possible to vary the K parameter spatially, for example, correlated with relief/gradient? This could be useful for sites such as the Nepal one where there is a large difference in relief along the profile. May be beyond the scope of the paper for the moment, but it would be interesting to see if there was a correlation between the required smoothing parameter and the surrounding relief, so you could preferentially smooth the profile in areas more prone to errors.

*Yes, it is possible to vary K spatially. As errors vary spatially as a function of hillslope gradients, it is a good idea to model K as a function of hillslope gradient. We mention this possibility in the last paragraph of the discussion. While we have not yet implemented optional spatially variable K in the software implemention of the crs algorithm, we may do so in the future.*

Page 7, Lines 24-28 and Appendix B: I downloaded TopoToolbox and implemented the CRS smoothing method (which was easy to do!). It may be useful in the Appendix to include a link to a tutorial for using the CRS method, which I saw exists on the TopoToolbox wordpress. The authors have provided the tool crsapp for visual checking of the parameters, which is great, but it would be useful to provide a tutorial for use of this tool as well.

*As mentioned by the reviewer, we have included the short tutorial on ksn values in our blog. There will certainly be more posts on this. Thus, we included a link to our blog (http://topotoolbox.wordpress.com) in the Appendix B.*

Figure 5: Could you expand upon this caption to show what is being represented here? Is this the offset for every pixel in the river profile?

*Yes, we have changed the caption:*

> *Fig. 5: Elevation offsets between different DEMs and benchmark LiDAR DEMs calculated for all river pixels for the Yakima and Big Tujunga dataset (see Fig. 4).*

Figure 7: It would be useful to see these data for the other two field sites as well as for the Nepal site, to show visually how the distribution of the residuals varies with relief.

*We have changed Figure 7 to include the other sites as well (see below). Note that we also chose to logarithmically scale the x-axis which better illustrates the tails of the distributions.*

[Figure]

*Fig. 3: Distributions of residuals derived from the CRS algorithm applied to different DEMs in the three study sites.*

Page 9, Section 6.2: I think including analysis of channel slope and curvature may be needed in order to compare the ability of the different DEM datasets to analyse topographic information. Although elevation may not vary much with grid resolution, parameters such as local slope and curvature have been shown to be very sensitive to grid resolution (e.g. Vaze et al., 2010 (Env. Modelling and Software); Grieve et al., 2016 (ESurf)), with the range of slope and curvature values decreasing with resolution.

*This is an issue that Reviewer 1 also addressed. Indeed, particular applications in tectonic geomorphology are often rather interested in along-river gradients than in elevations. To address this issue, our revised manuscript contains an additional section on how the CRS algorithm will affect river gradients. An analysis of data from the Yakima catchment shows that CRS is able to considerably narrow down the pixel-by-pixel differences between along-river gradients derived from a resampled LiDAR DEM and those obtained from CRS as compared to other methods.*

[Figure]

*Fig. 4: Comparison of pixel-by-pixel along-river gradients derived from benchmark LiDAR DEMs (resampled to 30 m) and global DEMs in the Yakima catchment. Negative gradients in the benchmark and unprocessed DEMs are due to erroneously increasing river elevations in downstream direction. The grey solid lines (thin lines: 10 and 90%iles, thick line: median) are derived from a quantile regression. The dashed line is a one-to-one reference line. The parameters of the CRS algorithm (tau and K) are those in Table 4 of the manuscript. Compared to the original profiles and those obtained by preprocessing techniques filling and carving, the CRS algorithm is able to reduce the differences between along-river gradients as indicated by the narrow interquantile (90-10%iles) range.*

Analysis of elevation values alone may suggest that the TanDEM-X dataset is not an improvement on the 30m datasets, but the higher resolution dataset may actually be more useful for extraction of these other metrics which are also important for channel profile analysis.

***We rewrote this sentence to emphasize the potentials of TanDEM-X DEMs.***

> ***The resolution of 12 m and high accuracy on hillslopes of the TanDEM-X DEM (Purinton and Bookhagen, 2017) will offer new opportunities, but whether and how TanDEM-X DEMs will offer any advantage for the analysis of river profiles in high mountain areas compared to previous DEMs with lower resolution needs further study.***

Page 10, Section 6.3: In general I like the title, and I get the idea of the good, bad, and ugly errors, but I think it could be expanded on a bit more - the authors should clarify here why random errors are good, and systematic errors are bad (can state more clearly that random errors are easier to smooth from the profile, whereas systematic errors are more difficult to distinguish from real signals).

***We have rewritten this section and avoided the terms good, bad, and ugly. Please also see the reply to a comment from Reviewer 1.***

Page 10, Section 6.4: I think some of this section makes more sense to have in the introduction to set the context of why developing the CRS algorithm is important.

*We added the following sentence at the end of the introduction:*

> *Our algorithms will aid the visual interpretation and automated analysis of longitudinal river profiles and has additional applications in hydrodynamic modelling.*

Technical corrections

Page 2, Lines 19-20: I'd suggest rewording this sentence - it's unclear at the moment. Do you mean that random errors may or may not be clustered spatially?

*We have reworded this sentence:*

> *"Random errors may also be autocorrelated, i.e., they cluster spatially (Oksanen and Sarjakoski, 2006)."*

Page 6, Line 15: rephrase sentence 'Thus derived profiles are monotonously decreasing downstream while filtering the wiggles'

*We don't really see how to rephrase the sentence.*

Page 9, Line 16-18: Split this into two sentences.

*Done as requested:*

[revised manuscript text omitted]
{1}{2} \boldsymbol{x}^T \boldsymbol{H} \boldsymbol{x} + \boldsymbol{f}^T \boldsymbol{x} \text{ such that } \begin{cases} \boldsymbol{A}\boldsymbol{x} \le \boldsymbol{b} \\ \boldsymbol{A}_{eq} \cdot \boldsymbol{x} = \boldsymbol{b}_{eq} \\ \boldsymbol{lb} \le \boldsymbol{x} \end{cases} \tag{A13}$$

where the notation is the same as in Appendix A3. The quadratic term in the problem is

$$\boldsymbol{H} = \begin{bmatrix} \boldsymbol{0} & \boldsymbol{0} & \boldsymbol{0} \\ \boldsymbol{0} & \boldsymbol{0} & \boldsymbol{0} \\ \boldsymbol{0} & \boldsymbol{0} & \boldsymbol{B} \end{bmatrix} \tag{A14}$$

[revised manuscript text omitted]

---

## Referee Report (RR1)

**Review of esurf-2017-50: 'Bumps in river profiles: uncertainty assessment and smoothing using quantile regression techniques' by W. Schwanghart and D. Scherler**

Fiona Clubb

October 30, 2017

The authors have done a great job of addressing the comments that were brought up in the first review, especially with the new analysis of channel gradients and the testing of the sensitivity of the CRS method to the K and $\tau$ parameters. I think the paper is now suitable for final publication following a couple of very minor technical corrections (see below).

Page 2, Line 3: should be 'DEMs' rather than 'DEM'.

Page 2, Line 20: should be 'have' rather than 'has'.

Page 10, Lines 0-8: It's interesting that the same set of parameter values doesn't minimise the error between the benchmark and smoothed profiles for both elevation and gradient. Based on this, it seems like the user should choose parameter values depending on which metrics they wish to use for their analysis. From Figure 11 it looks like the gradient optimisation smooths the profile more than the elevation optimisation for the Yakima catchment, is that correct? It would be good to state what the $K$ and $\tau$ parameters actually were for the Yakima catchment for the different optimisations (can probably do this on Figure 11 somewhere?)

Page 11, Lines 14-18: Related to the previous comment, I think it would be useful to add a sentence here discussing the fact that different $K$ values lead to differences between the errors between gradients and elevations, and that this is something users should take into account when calibrating the parameters for their specific use.

---

## Author Response (AR2)

Earth Surface Dynamics Discussions (esurf-2017-50)

**Bumps in river profiles: uncertainty assessment and smoothing using quantile regression techniques**

Wolfgang Schwanghart and Dirk Scherler

**Final response to comments by reviewers**

*Dear Editor,*

*we have addressed all technical issues raised by the reviewers. As requested, we covered the problem of different objective functions (optimization based on elevation or gradient) in the discussion (page 11, lines 16f). Moreover, we updated Fig. 11 as well as the caption to indicate the parameter values for which we derive minimal RMSE.*

*In addition, we have added the latest version of the software as supplement.*

*We thank both reviewers for their constructive comments*

[revised manuscript text omitted]